# Bayesian Network Structure Discovery Using Large Language Models

**Yinghuan Zhang**  *yinghuan.flash@gmail.com*
*Independent Researcher*

**Yufei Zhang**  *yufeizhang96@outlook.com*
*Independent Researcher*

**Parisa Kordjamshidi**  *kordjams@msu.edu*
*Department of Computer Science and Engineering*
*Michigan State University*

**Zijun Cui**  *cuizijun@msu.edu*
*Department of Computer Science and Engineering*
*Michigan State University*

**Reviewed on OpenReview:** *https://openreview.net/forum?id=G4mrO8LVix*

## Abstract

Understanding probabilistic dependencies among variables is central to analyzing complex systems. Traditional structure learning methods often require extensive observational data or are limited by manual, error-prone incorporation of expert knowledge. Recent studies have explored using large language models (LLMs) for structure learning, but most treat LLMs as auxiliary tools for pre-processing or post-processing, leaving the core learning process data-driven. In this work, we introduce a unified framework for Bayesian network structure discovery that places LLMs at the center, supporting both data-free and data-aware settings. In the data-free regime, we introduce **PromptBN**, which leverages LLM reasoning over variable metadata to generate a complete directed acyclic graph (DAG) in a single call. PromptBN effectively enforces global consistency and acyclicity through dual validation, achieving constant $\mathcal{O}(1)$ query complexity. When observational data are available, we introduce **ReActBN** to further refine the initial graph. ReActBN combines statistical evidence with LLM by integrating a novel ReAct-style reasoning with configurable structure scores (e.g., Bayesian Information Criterion). Experiments demonstrate that our method outperforms prior data-only, LLM-only, and hybrid baselines, particularly in low- or no-data regimes and on out-of-distribution datasets.

Code is available at `https://github.com/sherryzyh/llmbn`.

## 1 Introduction

Structure discovery of probabilistic graphical models (PGMs) is key to understanding complex systems, as it reveals the underlying dependency structure among variables. Traditional approaches, including constraint-based algorithms (Spirtes et al., 2001) and score-based methods (Chickering, 2002b), typically require large amounts of observational data and intensive computation, limiting their practical use. Incorporating domain knowledge is another option, but it is typically done manually, which is labor-intensive and error-prone. This motivates the development of new methods to infer probabilistic structures more effectively and efficiently.

Recently, large language models (LLMs) trained on large-scale text corpora have demonstrated remarkable performance (Brown, 2020), including capabilities in probabilistic reasoning and the extraction of structured

knowledge from text (Nafar et al., 2024; 2025a;b). Their ability to capture a broad range of dependencies, including implicit causal and statistical structures, makes LLMs a promising tool for structure learning. Several approaches attempt to use LLMs as sources of prior knowledge (Ban et al., 2025; Vashishtha et al., 2023; Long et al., 2023a) or as post-hoc verifiers (Khatibi et al., 2024). In both cases, the core structure learning process is constrained to be tightly coupled with data-driven optimization techniques. Directly eliciting full probabilistic graphs from LLMs remains challenging, mainly due to two factors: (1) the combinatorial complexity of graph construction, making it difficult for LLMs to generate globally coherent structures, and (2) the lack of effective mechanisms to enforce structural constraints (e.g., acyclicity) and to incorporate observational data within prompt-based interactions.

In this paper, we address these challenges by introducing a unified two-phase LLM-centered framework for Bayesian network (BN) structure learning. The first phase, PromptBN, directly elicits a coherent and acyclic directed acyclic graph (DAG) through a meta-prompting strategy with dual validation that operates entirely without observational data. When data are available, the second phase, ReActBN, performs iterative refinement in a Reason-and-Act (ReAct) workflow (Yao et al., 2023), where offline-computed structure scores (e.g., BIC) evaluate candidate neighbors and guide the model's decisions. Unlike prior LLM-based methods that delegate structural correction to external algorithms, our method maintains the LLM in the decision loop across both phases, enabling a more integrated and interpretable approach to BN discovery. Our key contributions are as follows:

- We propose an LLM-based framework for flexible structure discovery with or without observational data.

- We propose a meta-prompting approach that elicits a Directed Acyclic Graph (DAG) in one LLM inference with $\mathcal{O}(1)$ query complexity.

- We propose a novel agentic search framework to refine the initial structure by integrating Reason-and-Act (ReAct) for LLM internal knowledge reasoning with structure scores like BIC.

- We perform extensive experiments on diverse datasets, showing superior structure discovery performance over state-of-the-art methods. In specific, contamination tests show that the observed gains arise from reasoning rather than memorization. Complexity analyses further demonstrate scalability to large, complex graphs, especially compared with existing LLM-based approaches.

## 2 Related Work

**Data-driven structure learning**   Conventional structure learning methods infer Directed Acyclic Graphs (DAGs) primarily from observational data, broadly categorized into *constraint-based* and *score-based* approaches (Glymour et al., 2019; Scutari et al., 2019). Constraint-based methods, such as the Peter-Clark algorithm (Spirtes et al., 2001) and its variants, identify probabilistic structures by assessing conditional independencies. Score-based methods, including Hill Climbing (Tsamardinos et al., 2006) and Greedy Equivalence Search (Chickering, 2002a), optimize predefined scoring functions (e.g., Bayesian Information Criterion). Although these methods are effective with sufficient data, their performance deteriorates significantly with limited samples (Cui et al., 2022a). Researchers incorporate prior expert knowledge, such as structural constraints or topological ordering, into learning frameworks (Borboudakis & Tsamardinos, 2012; Constantinou et al., 2023; Chen et al., 2016) when data is insufficient. Eliciting expert knowledge, however, remains challenging and labor-intensive.

**LLM-guided structure discovery with data**   Recent methods explore LLMs in structure discovery, primarily as a complement to conventional approaches. Several researchers use pairwise prompting to guide LLMs in uncovering causal relationships between pairs of nodes (Choi et al., 2022; Kıcıman et al., 2023; Long et al., 2023b). On top of that, Jiralerspong et al. (2024) propose a Breadth-First Search (BFS) prompting approach, in which the LLM is prompted to find all connected nodes given the current node. Pearson correlation coefficients are also included in the prompts as hints.

In addition, LLMs have been used to provide structural priors, establish topological orderings, or refine learned probabilistic graphs post hoc (Vashishtha et al., 2023; Long et al., 2023a; Khatibi et al., 2024). Some methods also use LLM-generated statements to iteratively guide data-driven approaches (Ban et al., 2023; Chen et al., 2023) or to incorporate observational data into prompts under both pairwise and BFS paradigms for further improvement (Susanti & Färber, 2025). Beyond pairwise- and BFS-based prompting, Ban et al. (2025) introduced LLM-CD, a hybrid approach that treats the LLM as a structured knowledge retriever to elicit graph-level priors and structural constraints. Once these LLM-derived priors are established, a classical data-driven structure learning algorithm can be applied to conduct the subsequent search.

**LLM-only structure discovery without data** LLM-only structure learning has been explored in two main directions. Cohrs et al. (2024) propose an LLM-driven conditional independence (CI) testing procedure embedded within the PC framework. Although conceptually appealing, the repeated CI queries make the method computationally expensive and limit its applicability to small networks. In parallel, Babakov et al. (2024) investigates a multi-expert strategy in which a facilitator LLM generates multiple personas and aggregates their proposed causal relations. Although this design excels with $\mathcal{O}(1)$ query complexity effectively, it still admits the worst-case $\mathcal{O}(N^2)$ query complexity when resolving conflicting directed pairs.

These limitations leave open the need for a more efficient and integrative approach. Our method addresses this gap by directly eliciting a full causal graph through a single structured prompt—achieving $\mathcal{O}(1)$ query complexity—and by offering a unified framework that naturally extends to data-aware refinement when observational data are available.

## 3 Problem Definition

A Bayesian network (BN) $\mathcal{G}$ is a directed acyclic graph (DAG) consisting of nodes $\mathcal{V}$ and edges $\mathcal{E}$, i.e., $\mathcal{G} = (\mathcal{V}, \mathcal{E})$. The task of structure learning for BNs typically refers to learning the structure of a DAG from the data $\mathcal{D}$. For example, score-based structure learning can be formulated as a constrained optimization problem:

$$\mathcal{G}^* = \arg\max_{\mathcal{G} \in \mathbb{G}} \mathrm{Score}(\mathcal{G}, \mathcal{D})$$
$$\text{s.t. } \mathcal{G} \in \mathrm{DAG} \tag{1}$$

where $\mathcal{G}^*$ is the optimal network structure; $\mathbb{G}$ is the search space of all possible DAGs; $\mathrm{Score}(\mathcal{G}, \mathcal{D})$ is a scoring function that evaluates how well the structure $\mathcal{G}$ fits the data $\mathcal{D}$.

## 4 Proposed Approach

We have developed a two-phase framework for LLM-centered discovery of BN structures.

- Phase-1 **PromptBN**: A single-step prompting approach that elicits Bayesian network structures directly from variable metadata, relying exclusively on the LLM's internal knowledge base.

- Phase-2 **ReActBN**: A ReAct-inspired agent system that integrates score-based evaluation with LLM reasoning, where the LLM supervises and selects optimal actions from top-$k$ candidates identified through scoring mechanisms.

The framework overview can be found at Figure 1, and pseudocode is available at Appendix A.

### 4.1 PromptBN: Bayesian Network Estimation via Prompting

In Phase 1, we formulate structure learning as a single-query inference task:

$$\mathcal{G} = \text{LLM-Query}(metadata(\mathcal{V}); knowledge(\mathcal{M})),$$

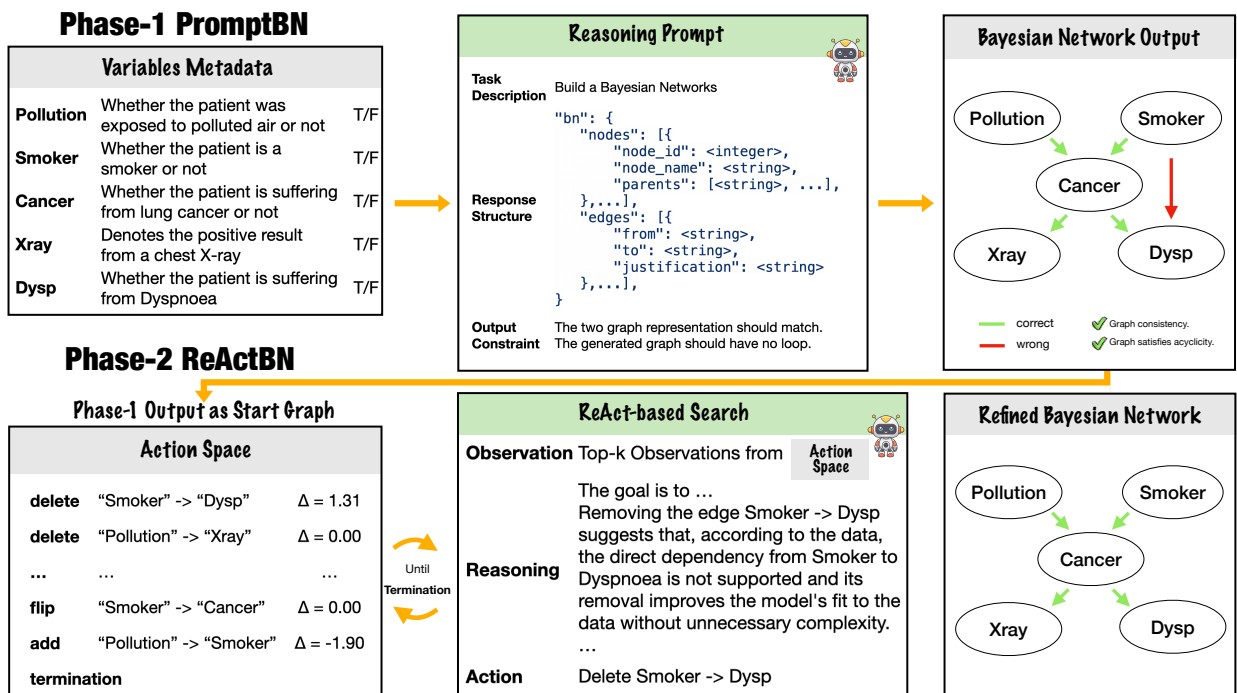

Figure 1: Overview of the proposed unified prompting framework for Bayesian network structure discovery using LLMs. The framework supports both data-free and data-driven settings through a two-phase process: Phase 1 induces the structure without observational data, while Phase 2 refines the estimated structure by incorporating data when available.

where LLM-Query$(\cdot)$ denotes a single query to the selected model $\mathcal{M}$, leveraging its implicit internal knowledge. Here, $\mathcal{V}$ represents the set of variables, and the *metadata* of each variable include its name, description of the variable, and distribution of the variable. An example input is illustrated in Figure 1.

Our approach employs meta-prompting (Zhang et al., 2023), where the prompt defines not only the task objective but also the reasoning protocol, response structure, and output constraints. Specifically, PromptBN requires the model to produce a valid Directed Acyclic Graph (DAG) while articulating its reasoning process.

To ensure robustness, we require the model to output the graph structure in two complementary representations: a node-centric format that specifies parent nodes for each variable, and an edge-centric format that enumerates all directed edges as "from-to" pairs.

Following generation, we apply dual validation procedures. First, *Structural Consistency Validation* confirms that parent–child relationships are identical across both representations. Second, *DAG Constraint Validation* verifies acyclicity. If the output fails either criterion, we prompt the LLM to regenerate the structure until validation succeeds or the maximum retry threshold is reached.

### 4.2 ReActBN: ReAct-based Score-Aware Search

As illustrated in Fig. 1, Phase 2 refines the initial Bayesian network (BN) structure obtained in Phase 1 by incorporating observational data. This phase adopts a ReAct-style (Reason + Act) iterative framework (Yao et al., 2023), referred to as **ReActBN**, in which an LLM explicitly reasons about candidate structural modifications and selects actions based on statistical feedback.

**Iterative Search Framework.** Starting from the current directed acyclic graph (DAG) $G_t$, ReActBN performs an iterative search procedure. At each reasoning iteration, ReActBN presents the LLM with the

top-$k$ candidate structures. These candidates are derived through an exhaustive enumeration of all valid neighboring structures, each annotated with its structure score and score differential.

**Structure Scoring.** The scoring function itself is fully configurable to accommodate different statistical criteria. Each candidate structure $G'$ is evaluated using a score-based criterion. In this work, we employ the Bayesian Information Criterion (BIC) defined as

$$\text{BIC}(G \mid D) = \log p(D \mid G, \hat{\theta}_G) - \frac{\dim(G)}{2} \log N, \tag{2}$$

where $D$ denotes the observational dataset, $\hat{\theta}_G$ are the maximum likelihood parameter estimates associated with $G$, $\dim(G)$ is the number of free parameters in the network, and $N$ is the sample size. For each candidate $G'$, we compute the score difference $\Delta_{G''} = \text{BIC}(G') - \text{BIC}(G_t)$.

**LLM-Guided Reasoning and Action Selection.** At each iteration, the LLM is provided with comprehensive contextual information, including variable metadata, the current graph structure $G_t$, its BIC score, the set of top-$k$ candidate actions with their associated score differences, and a tabu list of previously visited structures. Unlike traditional greedy score-based methods (e.g., Hill Climbing or Greedy Equivalence Search), which deterministically select the single highest-scoring move, ReActBN allows the LLM to reason over multiple candidate actions. The LLM outputs a selected action (or a termination decision), accompanied by an explicit reasoning trace and a confidence estimate. To encourage exploration and prevent cycling, we incorporate a tabu search mechanism that prohibits revisiting previously explored structures.

## 5 Experiments

In this section, we validate the effectiveness of our approach. We first describe the datasets, evaluation metrics, and experimental setup (Section 5.1). We then compare our method against existing approaches in both data-free and data-aware settings on classical (Section 5.2) and newer (Section 5.3) benchmarks, followed by a valuable study of potential contamination in LLMs (Section 5.4).

### 5.1 Evaluation Datasets and Metrics

**Datasets** We evaluate on ten datasets for Bayesian network structure learning. Specifically, seven datasets, including Asia, Cancer, Earthquake, Child, Insurance, Alarm, and Hailfinder, are widely used benchmarks in structure learning, from Bnlearn (Scutari, 2010). The remaining three datasets, blockchain, consequence-Covid (referred to covid hereafter), and disputed3, are recent datasets from BnRep (Leonelli, 2025). We retrieve all the metadata, including variable names, descriptions, and distributional properties, from the released packages, except for Insurance, whose description is taken from Long et al. (2023a). For the Earthquake dataset, since no quantitative results are available for comparison, we report qualitative evaluation results by visualizing the learned structure. We provide a summary of all dataset statistics in Appendix B.1.

**Evaluation metrics** We evaluate both the effectiveness and efficiency of structure learning algorithms. For effectiveness, we use two well-established metrics: **Structural Hamming Distance (SHD)** and **Normalized Hamming Distance (NHD)**. To compute SHD, both the predicted graph and the ground-truth DAG are first converted into their corresponding completed partially directed acyclic graphs (CPDAGs). SHD is then defined as the number of edge modifications, including insertions, deletions, or reversion, required to make the predicted CPDAG identical to the ground-truth CPDAG. NHD is a variant of SHD that offers a scale-invariant measure of structural difference. Following Khatibi et al. (2024), for a graph with $N$ nodes, NHD is calculated as $\sum_{i=1}^{N} \sum_{j=1}^{N} \frac{1}{N^2} \cdot \mathbf{1}(E_{ij} \neq \hat{E}_{ij})$, where $E_{ij}$ and $\hat{E}_{ij}$ denote the presence or absence of an edge from node $i$ to node $j$ in the ground truth and predicted graphs, respectively. $\mathbf{1}(\cdot)$ is the indicator function, which returns 1 if $E_{ij} \neq \hat{E}_{ij}$ and 0 otherwise.

**Experimental setup. 1) LLM models:** Unless otherwise specified, we use the OpenAI model `o3-2025-04-16` (referred to as o3 hereafter) for PromptBN and `gpt-4.1-2025-04-14` for ReActBN, accessed

via the OpenAI API (Achiam et al., 2023). We select o3 for PromptBN due to its strength in structured multi-step reasoning for single-shot DAG generation, and gpt-4.1 for ReActBN due to its reliable stepwise instruction following for iterative refinement. **2) Baselines:** We compare approaches across three categories: LLM, Data, and Hybrid. The LLM category contains methods that rely on large language models without using any observational data, including PairwisePrompt (Kıcıman et al., 2023), BFSPrompt (Jiralerspong et al., 2024), Scalability (Babakov et al., 2024), and ChatPC (Cohrs et al., 2024). The Data category consists of classical structural-learning algorithms that operate purely on observational samples through statistical estimation, including Hill Climbing (HC), Peter–Clark algorithm (PC-stable), Greedy Equivalence Search (GES), and three algorithms designed for low data observations (Cui et al., 2022b), i.e., RAI-BF, $rI^{eB}$, and $rBF_{chi2}$. The Hybrid category encompasses approaches that combine LLM reasoning with observational data, including LLM-CD (HC) (Ban et al., 2025) and PairwisePrompt/BFSPrompt+Data (Susanti & Färber, 2025). When reproducing results, we use the default parameters in the Python pgmpy package (Ankan & Textor, 2024) for HC, PC-stable, and GES. We use the same sample size for all LLM-based methods except ChatPC and Scalability, for which reproduction is not feasible. **3) Sample size:** For the data-aware setting, we use 100 samples following Cui et al. (2022a), unless otherwise specified. **4) Hyperparameters:** For ReActBN, the hyperparameters are listed in Table 8. **5) Repeated experiments and valid runs:** To address the intrinsic stochasticity of large language models, we report results over five valid runs, where a valid run is defined as one that successfully completes and produces a graph passing our dual-validation checks. We continue running experiments for each configuration until either: (1) five valid runs have been collected, or (2) five consecutive invalid runs occur, at which point we terminate that configuration.

## 5.2 Evaluation on Classical Benchmarks

We evaluate the performance of our proposed methods, **PromptBN** and **ReActBN**, and compare them with existing methods across the three categories of baselines: LLM, Data, and Hybrid approaches. We present the SHD and NHD evaluations on six widely used classical Bayesian network structure learning benchmarks (Asia, Cancer, Child, Insurance, Alarm, Hailfinder) in Table 1.

**Comparison to LLM-only methods (LLM)**   Among LLM-only methods for Bayesian network structure discovery (PairwisePrompt, BFSPrompt, and Scalability with GPT-4), our proposed **PromptBN** demonstrates the most robust and consistent performance. PromptBN achieves exact recovery on Asia (SHD = 0), high fidelity on Cancer (SHD = 0.6), and the top result on Insurance (SHD = 35.6). Even on larger networks such as Alarm and Hailfinder, PromptBN consistently attains either the best or second-best SHD and NHD. Furthermore, the performance of **PromptBN+HC** indicates that PromptBN provides substantially stronger initialization than an empty graph, improving downstream hill-climbing optimization across multiple datasets. For instance, on Hailfinder, PromptBN+HC reduces the SHD from 72 (HC alone) to 63.8.

As ChatPC (Cohrs et al., 2024) only reports qualitative evaluations without quantitative metrics, we visualize the structures produced by their method and ours in Figure 2 for comparison. On benchmarks such as Asia and Cancer, PromptBN yields noticeably cleaner and more faithful graph structures that exactly match the ground truth, underscoring the reliability of its global structure-generation mechanism.

**Comparison to Data-only Baselines (Data) and Hybrid Methods (Hybrid)**   As illustrated in Table 1, classical data-only algorithms, including PC-Stable, HC, and GES, typically achieve near-optimal accuracy on small networks but show rapid performance degradation on medium- and large-scale benchmarks such as Insurance and Hailfinder. Low data–oriented variants, such as RAI-BF, $rI^{eB}$, and $rBF_{chi2}$ (Cui et al., 2022b), may provide improvements on selected datasets but suffer from inconsistent performance overall. By contrast, hybrid methods achieves the enhanced performance when combining LLM outputs with data. Across all data-available settings, our proposed **ReActBN** achieves the strongest performance on nearly all classical benchmarks. ReActBN exactly recovers the Cancer network (SHD = 0) and attains the best results on 6 out of 9 datasets. On Insurance, ReActBN outperforms all baselines with an SHD of 40.2. On Hailfinder, it surpasses multiple specialized low-data algorithms, including $rI^{eB}$ and $rBF_{\chi^2}$, demonstrating its robustness in challenging high-dimensional scenarios.

Table 1: Evaluation on classical (Asia, Cancer, Child, Insurance, Alarm, Hailfinder) and newer (blockchain, covid, disputed3) Bayesian network datasets

| Category | #Sample | Algorithm | Asia | | Cancer | | Child | | Insurance | | Alarm | |
|---|---|---|---|---|---|---|---|---|---|---|---|---|
| | | | SHD↓ | NHD↓ | SHD↓ | NHD↓ | SHD↓ | NHD↓ | SHD↓ | NHD↓ | SHD↓ | NHD↓ |
| LLM | 0 | PairwisePrompt | 5.6 | 0.072 | **0.0** | **0.000** | 19.6 | 0.042 | 37.6 | 0.044 | 60.0 | 0.044 |
| LLM | 0 | BFSPrompt | **0.0** | **0.000** | **0.0** | **0.000** | 20.4 | 0.047 | 48.4 | 0.058 | 36.2 | 0.026 |
| LLM | 0 | Scalability (GPT-4)[1] | – | – | – | – | – | – | 52.0 | – | 60.0 | – |
| LLM | 0 | **PromptBN (Ours)** | **0.0** | **0.000** | 0.6 | 0.024 | 21.6 | 0.045 | **35.6** | **0.044** | 41.8 | 0.031 |
| Data | 100 | PC-Stable | 9.0 | 0.156 | 4.0 | 0.160 | 20.0 | 0.065 | 51.0 | 0.073 | 46.0 | 0.037 |
| Data | 100 | HC | 8.0 | 0.125 | 4.0 | 0.160 | 20.0 | 0.065 | 54.0 | 0.067 | 50.0 | 0.040 |
| Data | 100 | GES | 8.0 | 0.125 | 4.0 | 0.160 | 40.0 | 0.110 | 72.0 | 0.099 | 50.0 | 0.042 |
| Data | 100 | RAI-BF[2] | 5.7 | – | 4.1 | – | 30.4 | – | 54.9 | – | 48.4 | – |
| Data | 100 | $rI^{eB}$ [2] | 13.3 | – | 4.7 | – | 21.6 | – | 48.9 | – | 44.5 | – |
| Data | 100 | $rBF_{chi2}$ [2] | 7.7 | – | 4.1 | – | 24.2 | – | 50.1 | – | 42.7 | – |
| Hybrid | 100 | LLM-CD(HC) | **0.0** | **0.000** | **0.0** | **0.000** | 40.0 | 0.093 | 65.0 | 0.089 | **22.0** | **0.219** |
| Hybrid | 100 | PairwisePrompt+Data | 0.8 | 0.013 | **0.0** | **0.000** | 20.6 | 0.048 | 44.4 | 0.055 | 69.4 | 0.051 |
| Hybrid | 100 | BFSPrompt+Data | 0.4 | 0.006 | 0.2 | 0.008 | 24.4 | 0.054 | 50.4 | 0.061 | 42.0 | 0.030 |
| Hybrid | 100 | PromptBN+HC | 8.0 | 0.094 | **0.0** | **0.000** | 18.2 | 0.030 | 46.6 | 0.055 | 43.6 | 0.028 |
| Hybrid | 100 | PromptBN+RandomChoice | 5.6 | 0.084 | 1.2 | 0.056 | 23.4 | 0.053 | 43.4 | 0.056 | 43.0 | 0.031 |
| Hybrid | 100 | **ReActBN (Ours)** | 6.4 | 0.075 | **0.0** | **0.000** | **18.0** | **0.028** | 40.2 | 0.049 | 35.4 | 0.024 |

| Category | #Sample | Algorithm | Hailfinder | | blockchain | | covid | | disputed3 | |
|---|---|---|---|---|---|---|---|---|---|---|
| | | | SHD↓ | NHD↓ | SHD↓ | NHD↓ | SHD↓ | NHD↓ | SHD↓ | NHD↓ |
| LLM | 0 | PairwisePrompt | 551.0 | 0.176 | 37.4 | 0.260 | 82.6 | 0.433 | 41.2 | 0.057 |
| LLM | 0 | BFSPrompt | 243.0 | 0.077 | 29.0 | 0.204 | 74.6 | 0.401 | 21.6 | 0.027 |
| LLM | 0 | Scalability (GPT-4)[1] | 68.0 | – | – | – | – | – | – | – |
| LLM | 0 | **PromptBN (Ours)** | 76.8 | 0.025 | 15.2 | 0.100 | 45.0 | 0.225 | 15.6 | 0.018 |
| Data | 100 | PC-Stable | 76.0 | 0.024 | 13.0 | 0.104 | 36.0 | 0.160 | 35.0 | 0.069 |
| Data | 100 | HC | 72.0 | 0.024 | 13.2 | 0.094 | 32.6 | 0.156 | 30.8 | 0.051 |
| Data | 100 | GES | 155.0 | 0.050 | 13.0 | 0.083 | **29.0** | **0.142** | 33.0 | 0.053 |
| Data | 100 | RAI-BF[2] | – | – | 15.7 | – | 62.2 | – | 36.3 | – |
| Data | 100 | $rI^{eB}$ [2] | 88.0† | – | 14.6 | – | 47.2 | – | 34.7 | – |
| Data | 100 | $rBF_{chi2}$ [2] | 98.3† | – | 13.6 | – | 33.6 | – | 33.4 | – |
| Hybrid | 100 | LLM-CD(HC) | 150.0 | 0.045 | 12.0 | 0.090 | 34.0 | 0.169 | 16.0 | 0.026 |
| Hybrid | 100 | PairwisePrompt+Data | 356.0 | 0.114 | 27.2 | 0.188 | 69.4 | 0.362 | 35.6 | 0.047 |
| Hybrid | 100 | BFSPrompt+Data | 140.0 | 0.441 | 18.6 | 0.124 | 65.6 | 0.337 | 24.6 | 0.032 |
| Hybrid | 100 | PromptBN+HC | **63.8** | **0.018** | 12.2 | 0.085 | 33.2 | 0.156 | 15.4 | 0.019 |
| Hybrid | 100 | PromptBN+RandomChoice | 79.2 | 0.026 | – | – | – | – | – | – |
| Hybrid | 100 | **ReActBN (Ours)** | 75.0 | 0.023 | **11.0** | **0.074** | 32.8 | 0.164 | **12.2** | **0.015** |

[1] Reported in Babakov et al. (2024).
[2] Reported in Cui et al. (2022b).
†: the sample size is 500 instead of 100 for Hailfinder.

## 5.3 Evaluation on Newer Datasets

We evaluate on three recently introduced Bayesian networks (Leonelli, 2025), including blockchain, covid, and disputed3 to assess the generalization capability of our method. These datasets differ substantially from the widely used classical benchmarks and are far less likely to appear in model pre-training corpora. They therefore provide a meaningful proxy for out-of-distribution (OOD) evaluation. As no prior work reports results on these networks, we reproduce all baselines and summarize the results in Table 1. For fair comparison, the data-driven and hybrid baselines use 100 observational samples.

Across all three networks, both PromptBN and ReActBN demonstrate strong generalization performance. Under the data-free setting, PromptBN consistently achieves the best results, exhibiting a substantial performance improvements over existing LLM-only methods. For example, on covid, PromptBN attains an

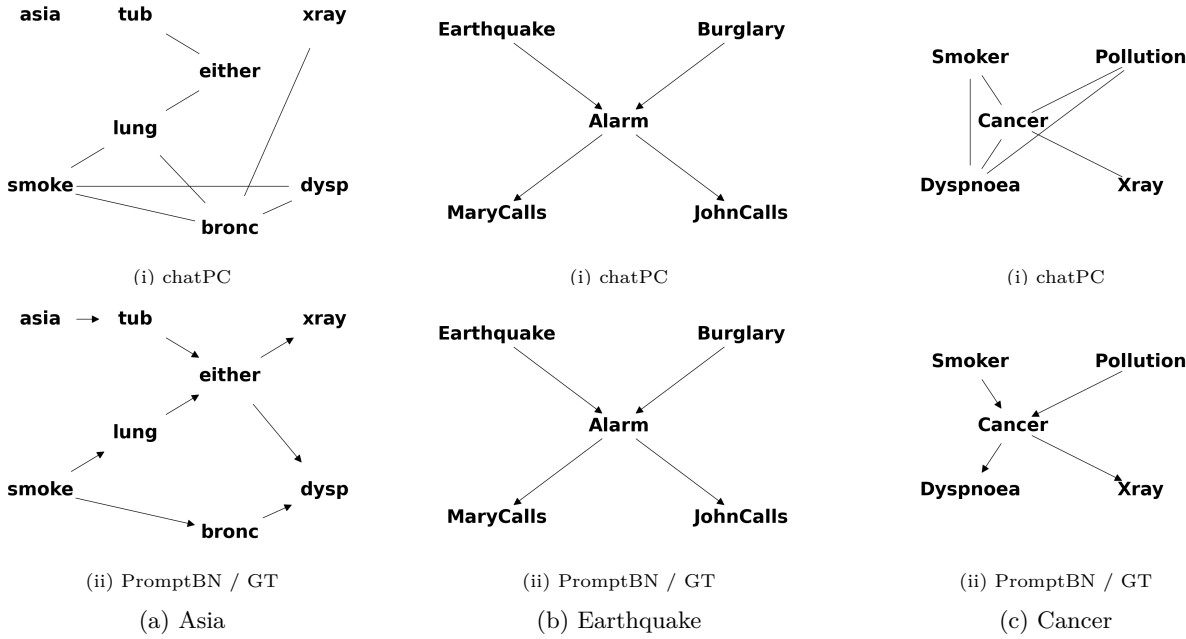

Figure 2: Qualitative comparison between our method (PromptBN) and chatPC across three benchmark datasets. Each group shows chatPC (top) and PromptBN/Ground Truth DAG (bottom): (a) Asia, (b) Earthquake, and (c) Cancer.

SHD of 45.0, outperforming PairwisePrompt (82.6) and BFSPrompt (74.6) by a wide margin. This high-lights PromptBN's ability to leverage metadata effectively without relying on any observational data, even in OOD scenarios. Moreover, under the data-aware setting, ReActBN achieves the strongest overall perfor-mance, obtaining the best results on two of the three networks (blockchain and disputed3), surpassing all LLM-only, data-only, and hybrid baselines. Even on covid, ReActBN remains competitive over LLM-based methods, showing its robustness under distribution shift and its advantage in balancing statistical evidence with LLM-driven structural reasoning. These findings demonstrate that PromptBN excels in zero-shot gen-eralization, while ReActBN provides state-of-the-art performance when even modest data is available. Both methods serve as strong and complementary solutions for structure learning in out-of-distribution regimes.

## 5.4 Contamination Tests and Analysis

Table 2: Contamination test with different forms of metadata input to LLMs

| Category | #Sample | Algorithm | Asia | | Cancer | | Child | | Insurance | | Alarm | | Hailfinder | |
|---|---|---|---|---|---|---|---|---|---|---|---|---|---|---|
| | | | SHD↓ | NHD↓ | SHD↓ | NHD↓ | SHD↓ | NHD↓ | SHD↓ | NHD↓ | SHD↓ | NHD↓ | SHD↓ | NHD↓ |
| LLM | 0 | VarName | 0.0 | 0.000 | 0.7 | 0.030 | 27.3 | 0.050 | 41.3 | 0.050 | 43.8 | **0.030** | 118.7 | 0.040 |
| LLM | 0 | VarDescription | 0.0 | 0.000 | **0.3** | **0.010** | 31.3 | 0.070 | 44.3 | 0.060 | 43.8 | **0.030** | 119.8 | 0.040 |
| LLM | 0 | ScrambleOrder | 0.0 | 0.000 | 0.6 | 0.024 | 24.8 | 0.051 | 37.4 | **0.044** | 48.4 | 0.964 | 97.8 | 0.032 |
| LLM | 0 | **PromptBN (Ours)** | **0.0** | **0.000** | 0.6 | 0.024 | **21.6** | **0.045** | **35.6** | **0.044** | **41.8** | 0.031 | **76.8** | **0.025** |

VarName: only variable names are provided
VarDescription: only variable descriptions are provided
ScrambleOrder: full metadata is given but in a randomly permuted order

Since the training corpora of modern LLMs are undisclosed, potential contamination cannot be fully ruled out. To better understand whether an LLM might recover underlying structures through memorization rather than genuine reasoning, we conduct three controlled contamination tests in which the LLM receives

a deliberately restricted or perturbed form of metadata and is asked to generate a Bayesian network using the same PromptBN instruction template. We present the experimental results in Table 2.

The contamination tests demonstrate that PromptBN's effectiveness is not attributable to memorization of dataset-specific patterns. When variable names (VarName) or descriptions (VarDescription) are used in isolation, structural recovery degrades sharply across all non-trivial networks—for example, Child (SHD 27.3–31.3), Insurance (41.3–44.3), Alarm (43.8), and Hailfinder (118.7–119.8). Such high errors indicate that the LLM cannot reconstruct the underlying causal structures from limited or incomplete metadata. These results provide strong evidence that PromptBN is not retrieving memorized graph templates, but is instead relying on genuine relational reasoning using the full set of metadata. Moreover, after permuting all variables and removing any positional or formatting cues (ScrambleOrder), the LLM can recover structures with SHD values that remain close to those of PromptBN—for instance, Child (24.8 vs. 21.6) and Insurance (37.4 vs. 35.6). Such stability under order perturbation confirms that PromptBN does not depend on memorizing any specific ordering of metadata for its predictions. Instead, it interprets the relational content itself, demonstrating robustness to ordering variations and reinforcing the idea that our method is performing global reasoning rather than exploiting data-specific patterns.

## 6 Complexity Analysis

In this section, we present a comprehensive computational complexity analysis. We first introduce the notation used throughout this section. We examine standard time complexity in classical methods and the offline CPU-side computational complexity in hybrid methods independent of any LLM calls (Section 6.1). We then provide a valuable study specific to LLMs for BN structure discovery, including query complexity (Section 6.2) and scalability (Section 6.3). We present an overview of complexities for our method and compare them to existing methods in Table 3.

**Notation** Let the BN structure learning problem be defined over $d$ random variables. We denote $N$ as the number of nodes (variables) in the Bayesian network, $E$ as the number of directed edges in the true or learned DAG, $i$ as the number of iterations (considered as a constant in the computational complexity analysis), and $k$ as the largest number of neighbors any variable has at any point during the PC adjacency removal process (i.e., the maximum adjacency size).

Table 3: Computational complexity analysis

| Algorithm | Category | Time Complexity | Query Complexity |
|---|---|---|---|
| Scalability | LLM | - | $\mathcal{O}(1)^*$ to $\mathcal{O}(N^2)$ |
| PairwisePrompt | LLM | - | $\mathcal{O}(N^2)$ |
| BFSPrompt | LLM | - | $\mathcal{O}(N)$ |
| ChatPC | LLM | - | $\mathcal{O}(N^2 \cdot 2^k)$ |
| **PromptBN (Ours)** | LLM | - | $\mathcal{O}(\mathbf{1})$ |
| PC-Stable | Data | $\mathcal{O}(N^2 \cdot 2^k)$ | - |
| HC | Data | $\mathcal{O}(N^2)$ | - |
| GES | Data | $\mathcal{O}(N^2)^*$ to $\mathcal{O}(N^3)$ | - |
| LLM-CD(HC) | Hybrid | $\mathcal{O}(N^2)$ | $\mathcal{O}(1)^*$ to $\mathcal{O}(N^2)$ |
| **ReActBN (Ours)** | Hybrid | $\mathcal{O}(N^2)$ | $\mathcal{O}(\mathbf{1})$ |

*: the value represents the best case depending on an internal LLM's generation step.

### 6.1 Time Complexity of Offline Computation

Both data-only algorithms and hybrid methods can incur substantial CPU overhead as the number of variables grows. We therefore compare classical data-only methods with the CPU components of hybrid approaches with full derivations, with complete complexity derivations in Appendix C.

PromptBN does not involve observational data and thus has no data-processing time complexity. ReActBN performs LLM-guided refinement with a classical search-and-score procedure, enumerating and evaluating local graph modifications at each iteration. Since each refinement step constructs the same $\mathcal{O}(N^2)$ neighborhood as HC, the dominant computation per iteration is $\mathcal{O}(N^2)$. With a bounded number of refinement iterations, the overall runtime of ReActBN scales as $\mathcal{O}(N^2)$. Compared to classical score-based and constraint-based algorithms (Data in Table 3), ReActBN introduces no additional computational overhead and exhibits similar complexity to SOTA hybrid methods such as LLM-CD when instantiated with the same score-based search component (i.e., HC). Nonetheless, ReActBN shows significantly lower query complexity than LLM-CD (HC), as discussed below.

## 6.2 LLM Query Complexity

LLM calls dominate both computational latency and monetary cost. We therefore study LLM query complexity, defined as the number of prompt-response interactions, to demonstrate our advantages in these two dimensions. We adopt the notion of **query complexity**—analogous to Big-$\mathcal{O}$ notation used in traditional time and space complexity analyses—to quantify the number of LLM API calls required by each algorithm. This metric applies specifically to LLM-based structural learning methods (LLM and Hybrid in Table 3). The complete query complexity derivations are in Appendix D.

PromptBN achieves constant query complexity of $\mathcal{O}(1)$ by generating the entire candidate DAG in a single LLM invocation, representing the optimal bound for any single-pass generative approach. In contrast, prior LLM-only methods require repeated pairwise or conditional-independence queries, leading to high query complexity. For example, ChatPC, which replaces the conditional independence (CI) tests in PC-Stable with LLM-based CI judgments, issues one query for each ordered pair of variables and for each conditioning set considered during adjacency removal. There are $\mathcal{O}(N^2)$ ordered variable pairs, and for each pair the algorithm enumerates up to $2^k$ conditioning subsets when the adjacency set has size at most $k$. The worst-case query complexity of ChatPC is $\mathcal{O}(N^2 2^k)$. By avoiding this bottleneck, PromptBN maintains competitive structural discovery accuracy while offering substantially lower query costs that would otherwise become prohibitive.

While further integrating LLM reasoning with classical score evaluations, ReActBN maintains query efficiency. Since the number of LLM queries equals the number of refinement iterations, which is fixed across datasets following standard Hill Climbing practice, ReActBN achieves query complexity of $\mathcal{O}(1)$ per iteration. This makes it substantially more query-efficient than existing hybrid methods, which require validation of all candidate edges and thus scale with network size. For example, LLM-CD introduces a three-stage prompting strategy. Its metadata derivation stage calls one query per variable, giving a cost of $\mathcal{O}(N)$. Then, the causal extraction stage invokes the LLM once to extract potential causal relationships. The final causal validation stage checks each extracted directed pair individually. In the worst case, the extraction stage can propose all $N(N-1)$ ordered pairs, resulting in $\mathcal{O}(N^2)$ validation queries, resulting in the overall worst-case query complexity $\mathcal{O}(N^2)$. The advantages of our approach in query complexity over existing LLM-only and hybrid methods further demonstrate its effectiveness.

## 6.3 Scalability

In practice, datasets vary considerably in size, making scalability to larger networks a critical consideration. We analyze how time and query complexity scale with dataset size and demonstrate both PromptBN and ReActBN are the only methods that (i) prevent LLM query counts from scaling with $N$ and (ii) maintain consistently strong performance across datasets, making them the most scalable solutions among all LLM-based methods considered.

Among LLM-only approaches (LLM), ChatPC is fundamentally unsuitable for moderate- or large-scale graphs. Its reliance on PC-Stable yields the poorest scalability, with query complexity mirroring the conditional-independence testing phase: $\mathcal{O}(N^2 \cdot 2^k)$ in the worst case, becoming exponential when the maximum adjacency size $k$ grows with $N$. In contrast, PromptBN achieves optimal scalability by generating the entire DAG in a single LLM inference, resulting in constant query complexity of $\mathcal{O}(1)$ and completely avoiding enumeration of conditioning sets or adjacency neighborhoods. PairwisePrompt and BFSPrompt fall

between these extremes, with quadratic and linear query growth respectively, but they remain significantly less scalable than PromptBN.

## 7 Factor Analysis

In this section, we study two fundamental factors when combining data with LLMs.

Table 4: Performance under different observational data sample sizes

| Category | #Sample | Algorithm | Asia | | Cancer | | Child | | Insurance | | Alarm | | Hailfinder | | blockchain | | covid | | disputed3 | |
|---|---|---|---|---|---|---|---|---|---|---|---|---|---|---|---|---|---|---|---|---|
| | | | SHD↓ | NHD↓ | SHD↓ | NHD↓ | SHD↓ | NHD↓ | SHD↓ | NHD↓ | SHD↓ | NHD↓ | SHD↓ | NHD↓ | SHD↓ | NHD↓ | SHD↓ | NHD↓ | SHD↓ | NHD↓ |
| Data | 100 | HC | 8.0 | 0.125 | 4.0 | 0.160 | 20.0 | 0.068 | 54.0 | 0.067 | 50.0 | 0.040 | 72.0 | 0.024 | 13.2 | 0.094 | **32.6** | **0.156** | 30.8 | 0.051 |
| Hybrid | 100 | PromptBN+HC | 8.0 | 0.094 | 0.0 | 0.000 | 18.2 | 0.030 | 46.6 | 0.055 | 43.6 | 0.028 | **63.8** | **0.018** | 12.2 | 0.085 | 33.2 | **0.156** | 15.4 | 0.019 |
| Hybrid | 100 | **ReActBN (Ours)** | **6.4** | **0.075** | **0.0** | **0.000** | 18.0 | 0.028 | 40.2 | 0.049 | 35.4 | 0.024 | 75.0 | 0.023 | **11.0** | **0.074** | 32.8 | 0.164 | **12.2** | **0.015** |
| Data | 250 | HC | 3.0 | 0.094 | 4.0 | 0.160 | 17.0 | 0.048 | 49.0 | 0.063 | 38.0 | 0.037 | 76.0 | 0.025 | 13.0 | 0.090 | 30.0 | 0.133 | 44.0 | 0.078 |
| Hybrid | 250 | PromptBN+HC | **1.0** | **0.016** | **0.0** | **0.000** | 13.2 | 0.026 | 37.6 | 0.046 | 32.8 | **0.026** | **65.8** | **0.019** | **9.6** | 0.079 | **28.4** | **0.140** | 11.8 | 0.013 |
| Hybrid | 250 | **ReActBN (Ours)** | **1.0** | **0.016** | **0.0** | **0.000** | **12.6** | 0.028 | **32.4** | **0.039** | **32.2** | **0.025** | 69.6 | 0.022 | 10.2 | **0.075** | 30.8 | 0.157 | **9.8** | **0.012** |

Table 5: Performance of PromptBN with different LLMs

| Model | Asia | | Cancer | | Child | | Insurance | | Alarm | | Hailfinder | | blockchain | | covid | | disputed3 | |
|---|---|---|---|---|---|---|---|---|---|---|---|---|---|---|---|---|---|---|
| | SHD↓ | NHD↓ | SHD↓ | NHD↓ | SHD↓ | NHD↓ | SHD↓ | NHD↓ | SHD↓ | NHD↓ | SHD↓ | NHD↓ | SHD↓ | NHD↓ | SHD↓ | NHD↓ | SHD↓ | NHD↓ |
| o3-pro | **0.0** | **0.000** | 1.0 | 0.040 | 22.2 | 0.045 | 37.6 | 0.041 | 44.8 | 0.033 | 79.6 | 0.026 | 12.8 | 0.085 | 44.2 | 0.226 | 9.8 | 0.010 |
| o3 | **0.0** | **0.000** | 0.6 | 0.024 | 21.6 | 0.044 | 35.6 | 0.043 | 41.8 | 0.030 | 76.8 | 0.057 | 15.2 | 0.045 | 45.0 | 0.224 | 15.6 | 0.018 |
| o4-mini | **0.0** | **0.000** | **0.0** | **0.000** | 22.8 | 0.050 | 43.2 | 0.053 | 46.8 | 0.035 | 81.8 | 0.055 | 13.2 | 0.042 | 42.8 | 0.234 | 19.6 | 0.024 |
| gpt-4.1 | **0.0** | **0.000** | **0.0** | **0.000** | **18.2** | **0.023** | 39.6 | 0.045 | 37.6 | 0.027 | 71.2 | 0.048 | 16.0 | 0.046 | 46.8 | 0.241 | 20.0 | 0.024 |
| gpt-4o | **0.0** | **0.000** | **0.0** | **0.000** | 22.0 | 0.040 | 46.2 | 0.054 | 32.0 | 0.021 | 64.0 | 0.020 | 14.4 | 0.094 | 44.0 | 0.229 | 26.4 | 0.034 |
| Deepseek-r1 | **0.0** | **0.000** | **0.0** | **0.000** | 18.2 | 0.038 | 48.8 | **0.037** | 48.8 | 0.037 | 65.4 | 0.021 | 13.6 | 0.044 | 44.2 | 0.237 | **5.2** | **0.007** |
| Deepseek-v3 | **0.0** | **0.000** | **0.0** | **0.000** | 21.2 | 0.046 | UnP | UnP | UnP | UnP | 65.8 | 0.021 | 16.2 | **0.041** | 41.4 | 0.207 | 28.2 | 0.033 |
| gemini-2.5-pro | **0.0** | **0.000** | **0.0** | **0.000** | 18.4 | 0.031 | **33.0** | 0.038 | **14.2** | **0.011** | 71.8 | 0.023 | **9.6** | 0.067 | 41.8 | 0.228 | 9.4 | 0.009 |
| gemini-2.0-flash | **0.0** | **0.000** | **0.0** | **0.000** | 21.2 | 0.045 | 47.4 | 0.064 | 40.4 | 0.029 | **54.4** | **0.017** | 14.0 | 0.094 | 39.8 | **0.192** | 28.0 | 0.038 |

[1] "UnP" (Unparsable) denotes cases where the LLM fails to strictly follow the output formatting requirements specified in the prompt, resulting in a response that cannot be parsed into a valid structure.

**Q1: How does the sample size of observational data affect performance?** Following widely adopted evaluation protocols (Cui et al., 2022a), we investigate how access to different amounts of observational data influences the performance of our framework. we compare our approach under two data regimes: 100 and 250 samples. For each setting, we evaluate three variants: (1) HC as a data-only baseline, (2) PromptBN+HC, which uses an LLM to initialize HC, and (3) ReActBN, which further leverages LLM reasoning to guide the score-based selection. As shown in Table 4, our proposed ReActBN achieves the best performance under sample sizes of both 100 and 250. For example, on Child, ReActBN achieves an SHD of 12.6, compared to 13.2 for PromptBN+HC and 17 for HC. Moreover, we note that the performance of conventional data-only methods improves as the amount of data increases, narrowing the gap with ReActBN. This trend is expected, as the reliance on LLM priors naturally decreases when sufficient data is available for reliable structure estimation.

**Q2: How does the choice of LLM affect structure learning performance?** We examine how different language models impact performance by evaluating PromptBN across a diverse set of LLMs, including proprietary models from OpenAI and Google and open-source models from DeepSeek. The tested models range from chat-oriented ones (e.g., DeepSeek-v3, GPT-4o, GPT-4.1) to reasoning-focused models (e.g., DeepSeek-r1, OpenAI o-series), and general-purpose language models (e.g., Gemini-2.5-Pro and Gemini-2.0-Flash). We present the results in Table 5. Interestingly, more advanced models with stronger general-purpose reasoning capabilities do not always achieve better performance in structure learning. For

instance, o3-pro, the strongest reasoning model from OpenAI, outperforms other models on most LLM benchmarking datasets. However, o3-pro performs worse than GPT-4o, the oldest model evaluated (o3-pro has an SHD of 44.8, while GPT-4o achieves 32.0). Similarly, Google's best model, Gemini-2.5-Pro, performs worse than Gemini-2.0-Flash on Hailfinder (with SHD of 71.8 vs. 54.4).

## 8 Conclusion

In this work, we present a unified, LLM-centered framework for Bayesian network structure discovery that operates effectively in both data-free and data-aware regimes. The first component, PromptBN, formulates structure learning as a single LLM query problem and produces a globally consistent DAG with constant query complexity $\mathcal{O}(1)$. The second component, ReActBN, integrates Reason-and-Act (ReAct)–based agentic reasoning with offline structural scores, such as BIC, to refine the initial graph in an LLM-as-brain process with also $\mathcal{O}(1)$ query complexity. Empirical evaluations across ten BN benchmarks demonstrate superior robustness, generalization, and low-data performance compared to state-of-the-art LLM-only, data-only, and hybrid baselines. Together, these results highlight the potential of LLMs as a central mechanism for scalable and principled probabilistic structure discovery.

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

# A  Pseudo Code and Examples

## A.1  PromptBN

Pseudo-code of the PromptBN (Phase-1 in our framework) is in Algorithm 1.

---

**Algorithm 1** PromptBN: Bayesian Network Estimation via Prompting

---

1: **Input:** Variable metadata table $\mathcal{I} = metadata(\mathcal{V})$, where $\mathcal{V}$ is the set of variables; LLM model $\mathcal{M}$; retry limit $N$
2: **Output:** Bayesian network $\mathcal{G}$
3: Construct meta-prompt $\mathcal{P}$ including:
4:     Task definition: instruct the LLM to infer a Bayesian network from $\mathcal{I}$
5:     Variable schema: include each variable's name, description, and distribution
6:     Output format: specify both node-centric and edge-centric representations
7:     Reasoning protocol: instruct the LLM to reason over variable metadata
8:     Response constraints: require strict formatting, completeness, and acyclicity
9: retry_count $\leftarrow 1$
10: **repeat**
11:     $\mathcal{O}_{raw} \leftarrow$ LLM-Query$(\mathcal{I}, \mathcal{P}; \mathcal{M})$
12:     Parse $\mathcal{O}_{raw}$ to obtain both:
13:         $\mathcal{G}_{node}$: node-centric representation
14:         $\mathcal{G}_{edge}$: edge-centric representation
15:     retry_count $\leftarrow$ retry_count $+1$
16: **until** (consistent$(\mathcal{G}_{node}, \mathcal{G}_{edge})$ **and** DAG$(\mathcal{G}_{edge})$) **or** retry_count $> N$
17: **if** retry_count $> N$ **then**
18:     **Raise error or halt:** Valid output not achieved within $N$ retries.
19: **else**
20:     **return** Bayesian network $\mathcal{G} \leftarrow \mathcal{G}_{edge}$
21: **end if**

---

## A.2  Input and Output Examples

An example of variable metadata $\mathcal{I}$ is shown in Table 6.

Table 6: Example input: $\mathcal{I}_{\text{Asia}} = metadata(\mathcal{V}_{\text{Asia}})$

| node | var_name | var_description | var_distribution |
|---|---|---|---|
| 1 | asia | visit to Asia | a two-level factor with levels *yes* and *no*. |
| 2 | tub | tuberculosis | a two-level factor with levels *yes* and *no*. |
| 3 | smoke | smoking | a two-level factor with levels *yes* and *no*. |
| 4 | lung | lung cancer | a two-level factor with levels *yes* and *no*. |
| 5 | bronc | bronchitis | a two-level factor with levels *yes* and *no*. |
| 6 | either | tuberculosis versus lung cancer/bronchitis | a two-level factor with levels *yes* and *no*. |
| 7 | xray | chest X-ray | a two-level factor with levels *yes* and *no*. |
| 8 | dysp | dyspnoea | a two-level factor with levels *yes* and *no*. |

The following is an example an LLM response for Asia dataset, illustrating the defined structure query.

```
"bn": {
    "nodes": [
        ...,
        {
            "node_id": 8,
            "node_name": "dysp",
```

```
        "parents": ["bronc", "either"],
        "description": "dyspnoea",
        "distribution": "a two-level factor with levels yes and no.",
        "conditional_probability_table": "P(dysp|bronc, either)"
    }
],
"edges": [
    ...,
    {
        "from": "bronc",
        "to": "dysp",
        "justification": "Bronchitis can cause dyspnoea."
    },
    {
        "from": "either",
        "to": "dysp",
        "justification": "Either tuberculosis or lung cancer can cause dyspnoea."
    }
],
"network_summary": "The Bayesian Network models the relationships between visiting Asia, smoking, and respiratory
diseases such as tuberculosis, lung cancer, and bronchitis. It captures how these factors influence symptoms like
dyspnoea and diagnostic results like chest X-rays."
}
```

### A.3 ReActBN

Pseudo-code of the ReActBN (Phase-2 in our framework) is in Algorithm 2.

## B Experiments

### B.1 Dataset details

Table 7: Statistics of the datasets used for method validation. † covid is short for consequenceCovid.

| Dataset | Asia | Cancer | Earthquake | Child | Insurance | Alarm | Hailfinder | blockchain | covid† | disputed3 |
|---|---|---|---|---|---|---|---|---|---|---|
| #Nodes | 8 | 5 | 5 | 20 | 27 | 37 | 56 | 12 | 15 | 27 |
| #Edges | 8 | 4 | 4 | 25 | 52 | 46 | 66 | 13 | 34 | 34 |

### B.2 Hyperparameters

Table 8: Summary of key hyperparameters.

| Hyperparameter | Meaning | Value |
|---|---|---|
| top-k | the candidate action space size | 10 |
| max_iter | the maximum number of iterations | 20 |
| tabu_length | the size of the tabu list | 100 |
| scoring | the type of structure score | BIC |

### B.3 Variance Statistics

In all our experiments, we report the mean and standard deviation with 5 valid runs under the same settings. To complement the results in the main article, we report the **standard deviation** in the following tables to provide a clearer view of the stability and robustness of our method:

---

**Algorithm 2** ReActBN: ReAct-based Score-Aware Search

---

**Require:** Initial Bayesian network $\mathcal{G}_0$; variable metadata $\mathcal{I}$; observational data $\mathcal{D}$
**Require:** Hyperparameters: top-$k$ $k$; scoring method $\mathcal{S}$; LLM model $\mathcal{M}$; tabu list length $L$; maximum iterations $T$
**Ensure:** Refined Bayesian network $\mathcal{G}^*$

1: $\mathcal{G} \leftarrow \mathcal{G}_0$; $s \leftarrow \mathcal{S}(\mathcal{G})$; $TL \leftarrow \emptyset$; $t \leftarrow 1$
2: **while** $t \leq T$ **do**
3:     $\mathcal{A} \leftarrow \emptyset$
4:     **for all** ordered pairs of nodes $(u, v)$ **do**
5:         **for all** actions $a \in \{\text{ADD}, \text{REMOVE}, \text{FLIP}\}$ **do**
6:             **if** $a$ is applicable to $(u, v)$ and applying $a$ keeps the graph acyclic **then**
7:                 $\mathcal{N} \leftarrow \mathcal{G} + a(u, v)$
8:                 **if** $\mathcal{N} \notin TL$ **then**
9:                     $\mathcal{A} \leftarrow \mathcal{A} \cup \{a(u, v)\}$
10:                     $s_{a(u,v)} \leftarrow \mathcal{S}(\mathcal{N})$
11:                     $\Delta_{a(u,v)} \leftarrow s_{a(u,v)} - s$
12:                 **end if**
13:             **end if**
14:         **end for**
15:     **end for**
16:     $\mathcal{A}_{\text{cand}} \leftarrow$ top-$k$ elements of $\mathcal{A}$ with the largest $\Delta_{a(u,v)}$
17:     Build prompt $\mathcal{P}$ using $(\mathcal{I}, \mathcal{G}, s, \mathcal{A}_{\text{cand}})$
18:     $idx \leftarrow \text{LLMQUERY}(\mathcal{P}; \mathcal{M})$                       $\triangleright$ $idx \in \{-1, 0, \ldots, k-1\}$
19:     **if** $idx = -1$ **then**
20:         **break**
21:     **else**
22:         $a^\star \leftarrow \mathcal{A}_{\text{cand}}[idx]$
23:         $\mathcal{G}_{\text{new}} \leftarrow \mathcal{G} + a^\star$
24:         **if not** $\text{acyclic}(\mathcal{G}_{\text{new}})$ **then**
25:             **continue**
26:         **end if**
27:         $TL \leftarrow TL \cup \{\mathcal{G}_{\text{new}}\}$
28:         **if** $|TL| > L$ **then**
29:             remove oldest element from $TL$
30:         **end if**
31:         $\mathcal{G} \leftarrow \mathcal{G}_{\text{new}}$
32:         $s \leftarrow \mathcal{S}(\mathcal{G})$
33:     **end if**
34:     $t \leftarrow t + 1$
35: **end while**
36: **return** $\mathcal{G}^* \leftarrow \mathcal{G}$

---

- Table 9 reports the variance of LLM-only and hybrid experiment, complementing Table 1.

- Table 10 reports the results of Table 4 in the main article.

- Table 11 reports the results of Table 5 in the main article.

## C  Detailed Time Complexity Analysis

Classical data-driven structure learning algorithms—including score-based approaches such as Hill Climbing (HC) and Greedy Equivalence Search (GES), as well as constraint-based approaches such as PC-Stable—exhibit distinct computational characteristics.

Table 9: LLM-only and Hybrid Evaluation on classical (Asia, Cancer, Child, Insurance, Alarm, Hailfinder) and newer (blockchain, covid, disputed3) Bayesian network datasets with Variance

| Category | # Sample | Algorithm | Asia | | Cancer | | Child | | Insurance | | Alarm | |
|---|---|---|---|---|---|---|---|---|---|---|---|---|
| | | | SHD↓ | NHD↓ | SHD↓ | NHD↓ | SHD↓ | NHD↓ | SHD↓ | NHD↓ | SHD↓ | NHD↓ |
| LLM | 0 | PairwisePrompt | 5.6 ± 0.5 | 0.072 ± 0.018 | 0.0 ± 0.0 | 0.000 ± 0.000 | 19.6 ± 3.1 | 0.042 ± 0.007 | 37.6 ± 3.3 | 0.044 ± 0.002 | 60.0 ± 14.3 | 0.044 ± 0.012 |
| LLM | 0 | BFSPrompt | 0.0 ± 0.0 | 0.000 ± 0.000 | 0.0 ± 0.0 | 0.000 ± 0.000 | 20.4 ± 2.1 | 0.047 ± 0.004 | 48.4 ± 13.3 | 0.058 ± 0.015 | 36.2 ± 2.8 | 0.026 ± 0.002 |
| LLM | 0 | Scalability (GPT-4)[1] | – | – | – | – | – | – | 52.0 | – | 60.0 | – |
| LLM | 0 | **PromptBN (Ours)** | **0.0 ± 0.0** | **0.000 ± 0.000** | **0.6 ± 0.6** | **0.024 ± 0.022** | 21.6 ± 5.7 | 0.044 ± 0.012 | **35.6 ± 3.8** | **0.044 ± 0.004** | 41.8 ± 5.2 | 0.031 ± 0.004 |
| Hybrid | 100 | LLM-CD(HC) | **0.0** | **0.000** | **0.0** | **0.000** | 40.0 | 0.093 | 65.0 | 0.089 | **22.0** | **0.219** |
| Hybrid | 100 | PairwisePrompt+Data | 0.8 ± 1.3 | 0.013 ± 0.020 | 0.0 ± 0.0 | 0.000 ± 0.000 | 20.6 ± 4.3 | 0.048 ± 0.011 | 44.4 ± 3.0 | 0.055 ± 0.004 | 69.4 ± 3.8 | 0.051 ± 0.003 |
| Hybrid | 100 | BFSPrompt+Data | 0.006 ± 0.014 | 0.2 ± 0.4 | 0.008 ± 0.018 | 24.4 ± 5.9 | 0.054 ± 0.014 | 50.4 ± 8.9 | 0.061 ± 0.008 | 42.0 ± 7.6 | 0.030 ± 0.005 | |
| Hybrid | 100 | **PromptBN+HC** | 8.0 ± 0.0 | 0.094 ± 0.006 | 0.0 ± 0.0 | 0.000 ± 0.000 | 18.2 ± 1.1 | 0.030 ± 0.003 | 46.6 ± 2.4 | 0.055 ± 0.006 | 43.6 ± 0.6 | 0.028 ± 0.001 |
| Hybrid | 100 | PromptBN+RandomChoice | 5.6 | 0.084 | 1.2 | 0.056 | 23.4 | 0.053 | 43.4 | 0.056 | 43.0 | 0.031 |
| Hybrid | 100 | **ReActBN (Ours)** | 6.4 ± 2.2 | 0.075 ± 0.026 | **0.0 ± 0.0** | **0.000 ± 0.000** | **18.0 ± 1.6** | **0.028 ± 0.004** | 40.2 ± 4.5 | 0.049 ± 0.004 | 35.4 ± 3.4 | 0.024 ± 0.001 |

| Category | # Sample | Algorithm | Hailfinder | | blockchain | | covid | | disputed3 | |
|---|---|---|---|---|---|---|---|---|---|---|
| | | | SHD↓ | NHD↓ | SHD↓ | NHD↓ | SHD↓ | NHD↓ | SHD↓ | NHD↓ |
| LLM | 0 | PairwisePrompt | 551.0 ± 18.9 | 0.176 ± 0.006 | 37.4 ± 2.3 | 0.260 ± 0.016 | 82.6 ± 2.5 | 0.433 ± 0.024 | 41.2 ± 4.8 | 0.057 ± 0.007 |
| LLM | 0 | BFSPrompt | 243.0 ± 52.9 | 0.077 ± 0.017 | 29.0 ± 6.8 | 0.204 ± 0.052 | 74.6 ± 8.6 | 0.402 ± 0.058 | 21.6 ± 7.1 | 0.027 ± 0.011 |
| LLM | 0 | Scalability (GPT-4)[1] | 68.0 | – | | | | | | |
| LLM | 0 | **PromptBN (Ours)** | 76.8 ± 6.7 | 0.025 ± 0.003 | 15.2 | 0.100 | 45.0 | 0.225 | 15.6 | 0.018 |
| Hybrid | 100 | LLM-CD(HC) | 150.0 | 0.045 | 12.0 | 0.090 | 34.0 | 0.169 | 16.0 | 0.026 |
| Hybrid | 100 | PairwisePrompt+Data | 356.0 ± 7.5 | 0.114 ± 0.003 | 27.2 ± 1.3 | 0.188 ± 0.011 | 69.4 ± 3.4 | 0.362 ± 0.013 | 35.6 ± 2.9 | 0.047 ± 0.004 |
| Hybrid | 100 | BFSPrompt+Data | 140.0 ± 52.9 | 0.044 ± 0.017 | 18.6 ± 3.7 | 0.124 ± 0.030 | 65.6 ± 5.0 | 0.337 ± 0.034 | 24.6 ± 4.5 | 0.032 ± 0.004 |
| Hybrid | 100 | PromptBN+HC | **63.8 ± 0.5** | **0.018 ± 0.000** | 12.2 | 0.085 | 33.2 | 0.156 | 15.4 | 0.019 |
| Hybrid | 100 | PromptBN+RandomChoice | 79.2 | 0.026 | – | – | – | – | – | – |
| Hybrid | 100 | **ReActBN (Ours)** | 75.0 ± 3.9 | 0.023 ± 0.002 | **11.0** | **0.074** | 32.8 | 0.164 | **12.2** | **0.015** |

[1] Reported in Babakov et al. (2024).
[2] Reported in Cui et al. (2022b).
[†]: the sample size is 500 instead of 100 for Hailfinder.

Table 10: Performance under different observational data sample sizes with variance

| Category | # Sample | Algorithm | Asia | | Cancer | | Child | | Insurance | | Alarm | |
|---|---|---|---|---|---|---|---|---|---|---|---|---|
| | | | SHD↓ | NHD↓ | SHD↓ | NHD↓ | SHD↓ | NHD↓ | SHD↓ | NHD↓ | SHD↓ | NHD↓ |
| Data | 100 | HC | 8.0 | 0.125 | 4.0 | 0.160 | 20.0 | 0.068 | 54.0 | 0.067 | 50.0 | 0.040 |
| LLM+Data | 100 | PromptBN+HC | 8.0±0.0 | 0.094±0.000 | 0.0±0.0 | 0.000±0.000 | 18.2±1.1 | 0.030±0.003 | 46.6±2.41 | 0.055±0.006 | 43.6±0.55 | 0.028±0.001 |
| LLM+Data | 100 | **ReActBN (Ours)** | 6.4±2.2 | 0.075±0.026 | 0.0±0.0 | 0.000±0.000 | 18.0±1.58 | 0.028±0.004 | 40.2±4.49 | 0.049±0.004 | 35.4±3.44 | 0.024±0.001 |
| Data | 250 | HC | 3.0 | 0.094 | 4.0 | 0.160 | 17.0 | 0.048 | 49.0 | 0.063 | 38.0 | 0.037 |
| LLM+Data | 250 | PromptBN+HC | 1.0±0.0 | 0.016±0.000 | 0.0±0.0 | 0.000±0.000 | 13.2±1.10 | 0.026±0.004 | 37.6±3.36 | 0.046±0.005 | 32.8±3.63 | 0.026±0.003 |
| LLM+Data | 250 | **ReActBN (Ours)** | 1.0±0.0 | 0.016±0.000 | 0.0±0.0 | 0.000±0.000 | 12.6±1.34 | 0.028±0.006 | 32.4±4.16 | 0.039±0.002 | 32.2±2.49 | 0.025±0.003 |

| Category | # Sample | Algorithm | Hailfinder | | blockchain | | covid | | disputed3 | |
|---|---|---|---|---|---|---|---|---|---|---|
| | | | SHD↓ | NHD↓ | SHD↓ | NHD↓ | SHD↓ | NHD↓ | SHD↓ | NHD↓ |
| Data | 100 | HC | 72.0 | 0.024 | 13.2 | 0.094 | **32.6** | **0.156** | 30.8 | 0.051 |
| LLM+Data | 100 | PromptBN+HC | **63.8±0.45** | **0.018±0.000** | 12.2±1.64 | 0.085±0.022 | 33.2±2.49 | 0.156±0.008 | 15.4±4.34 | 0.019±0.006 |
| LLM+Data | 100 | **ReActBN (Ours)** | 75.0±3.87 | 0.023±0.002 | **11.0±0.00** | **0.074±0.006** | 32.8±2.49 | 0.164±0.014 | **12.2±2.86** | **0.015±0.005** |
| Data | 250 | HC | 76.0 | 0.025 | 13.0 | 0.090 | 30.0 | 0.133 | 44.0 | 0.078 |
| LLM+Data | 250 | PromptBN+HC | **65.8±0.84** | **0.019±0.001** | 9.6±1.34 | 0.079±0.006 | **28.4±1.95** | **0.140±0.004** | 11.8±7.36 | 0.013±0.006 |
| LLM+Data | 250 | **ReActBN (Ours)** | 69.6±1.52 | 0.022±0.001 | 10.2±1.64 | 0.075±0.003 | 30.8±1.30 | 0.157±0.014 | **9.8±5.85** | **0.012±0.007** |

**Hill Climbing (HC)** as a local search method, HC evaluates the neighborhood $\mathcal{N}(\mathcal{G})$ of each candidate DAG, whose size is $\mathcal{O}(N^2)$ under standard add, delete, and reverse operators. With $i$ search iterations, the worst-case runtime of HC is therefore $\mathcal{O}(i \cdot N^2)$, consistent with empirical findings that its execution time grows approximately quadratically in $N$ (Liu et al., 2022). Since in practice the number of iterations is usually a constant across datasets, we keep the time complexity of HC at $\mathcal{O}(N^2)$.

**Greedy Equivalence Search (GES)** GES performs a two-phase (forward and backward) greedy search over equivalence classes. Each operator evaluation requires computing score differences for the parent sets affected by the modification. As observed in the FastGES (Ramsey et al., 2016), the per-phase runtime scales practically as $\mathcal{O}(N^2)$, though the theoretical upper bound may reach $\mathcal{O}(N^3)$ when many admissible covered-edge reversals are considered. Thus, the overall time complexity of GES ranges between $\mathcal{O}(N^2)$ and $\mathcal{O}(N^3)$ depending on structural density and operator availability.

**PC-Stable** PC-Stable exhibits the most computationally intensive profile among the classical methods due to its reliance on conditional independence (CI) tests as conditioning-set sizes increase. Its runtime

Table 11: Performance of PromptBN with different LLMs with variance

| Model | Asia SHD↓ | Asia NHD↓ | Cancer SHD↓ | Cancer NHD↓ | Child SHD↓ | Child NHD↓ | Insurance SHD↓ | Insurance NHD↓ | Alarm SHD↓ | Alarm NHD↓ |
|---|---|---|---|---|---|---|---|---|---|---|
| o3-pro | $0.0 \pm 0.0$ | $0.000 \pm 0.000$ | $1.0 \pm 0.0$ | $0.040 \pm 0.000$ | $22.2 \pm 4.9$ | $0.045 \pm 0.011$ | $37.6 \pm 2.6$ | $0.041 \pm 0.005$ | $44.8 \pm 5.3$ | $0.033 \pm 0.004$ |
| o3 | $0.0 \pm 0.0$ | $0.000 \pm 0.000$ | $0.6 \pm 0.6$ | $0.024 \pm 0.022$ | $21.6 \pm 5.7$ | $0.044 \pm 0.012$ | $35.6 \pm 3.8$ | $0.044 \pm 0.004$ | $41.8 \pm 5.2$ | $0.031 \pm 0.004$ |
| o4-mini | $0.0 \pm 0.0$ | $0.000 \pm 0.000$ | $0.0 \pm 0.0$ | $0.000 \pm 0.000$ | $22.8 \pm 5.1$ | $0.050 \pm 0.011$ | $43.2 \pm 4.3$ | $0.053 \pm 0.007$ | $46.8 \pm 8.2$ | $0.035 \pm 0.006$ |
| gpt-4.1 | $0.0 \pm 0.0$ | $0.000 \pm 0.000$ | $0.0 \pm 0.0$ | $0.000 \pm 0.000$ | $\mathbf{18.2 \pm 1.6}$ | $\mathbf{0.023 \pm 0.004}$ | $39.6 \pm 2.7$ | $0.045 \pm 0.002$ | $\mathbf{37.6 \pm 3.0}$ | $\mathbf{0.027 \pm 0.002}$ |
| gpt-4o | $0.0 \pm 0.0$ | $0.000 \pm 0.000$ | $0.0 \pm 0.0$ | $0.000 \pm 0.000$ | $22.0 \pm 0.0$ | $0.040 \pm 0.000$ | $46.2 \pm 0.4$ | $0.054 \pm 0.002$ | $32.0 \pm 6.1$ | $0.021 \pm 0.003$ |
| Deepseek-r1 | $0.0 \pm 0.0$ | $0.000 \pm 0.000$ | $0.0 \pm 0.0$ | $0.000 \pm 0.000$ | $18.2 \pm 4.1$ | $0.038 \pm 0.013$ | $\mathbf{35.4 \pm 7.0}$ | $\mathbf{0.042 \pm 0.008}$ | $48.8 \pm 7.6$ | $0.037 \pm 0.009$ |
| Deepseek-v3 | $0.0 \pm 0.0$ | $0.000 \pm 0.000$ | $0.0 \pm 0.0$ | $0.000 \pm 0.000$ | $21.2 \pm 0.5$ | $0.046 \pm 0.002$ | UnP | UnP | UnP | UnP |
| gemini-2.5-pro | $0.0 \pm 0.0$ | $0.000 \pm 0.000$ | $0.0 \pm 0.0$ | $0.000 \pm 0.000$ | $18.4 \pm 3.7$ | $0.031 \pm 0.011$ | $33.0 \pm 3.7$ | $0.038 \pm 0.004$ | $\mathbf{14.2 \pm 6.3}$ | $\mathbf{0.011 \pm 0.004}$ |
| gemini-2.0-flash | $0.0 \pm 0.0$ | $0.000 \pm 0.000$ | $0.0 \pm 0.0$ | $0.000 \pm 0.000$ | $21.2 \pm 1.9$ | $0.045 \pm 0.006$ | $47.4 \pm 3.7$ | $0.064 \pm 0.006$ | $40.4 \pm 3.3$ | $0.029 \pm 0.002$ |

| Model | Hailfinder SHD↓ | Hailfinder NHD↓ | blockchain SHD↓ | blockchain NHD↓ | consequenceCovid SHD↓ | consequenceCovid NHD↓ | disputed3 SHD↓ | disputed3 NHD↓ |
|---|---|---|---|---|---|---|---|---|
| o3-pro | $79.6 \pm 11.7$ | $0.026 \pm 0.004$ | $12.8 \pm 4.0$ | $0.085 \pm 0.020$ | $44.2 \pm 2.8$ | $0.226 \pm 0.011$ | $\mathbf{9.8 \pm 2.8}$ | $\mathbf{0.010 \pm 0.002}$ |
| o3 | $76.8 \pm 6.7$ | $0.025 \pm 0.003$ | $15.2 \pm 6.5$ | $0.100 \pm 0.040$ | $45.0 \pm 2.8$ | $0.225 \pm 0.017$ | $15.6 \pm 2.2$ | $0.018 \pm 0.004$ |
| o4-mini | $81.8 \pm 10.8$ | $0.027 \pm 0.004$ | $13.2 \pm 1.9$ | $0.092 \pm 0.013$ | $42.8 \pm 3.0$ | $0.234 \pm 0.013$ | $19.6 \pm 5.3$ | $0.024 \pm 0.008$ |
| gpt-4.1 | $\mathbf{64.0 \pm 6.8}$ | $\mathbf{0.020 \pm 0.002}$ | $16.0 \pm 0.0$ | $0.111 \pm 0.000$ | $46.8 \pm 1.5$ | $0.242 \pm 0.007$ | $20.0 \pm 4.9$ | $0.025 \pm 0.006$ |
| gpt-4o | $\mathbf{64.0 \pm 0.0}$ | $\mathbf{0.020 \pm 0.000}$ | $14.4 \pm 0.6$ | $\mathbf{0.107 \pm 0.004}$ | $44.0 \pm 1.2$ | $0.229 \pm 0.007$ | $26.4 \pm 1.7$ | $0.034 \pm 0.003$ |
| Deepseek-r1 | $65.4 \pm 6.3$ | $0.021 \pm 0.002$ | $\mathbf{13.6 \pm 3.7}$ | $0.094 \pm 0.026$ | $\mathbf{44.2 \pm 5.2}$ | $\mathbf{0.237 \pm 0.024}$ | $\mathbf{5.2 \pm 2.2}$ | $\mathbf{0.007 \pm 0.003}$ |
| Deepseek-v3 | UnP | UnP | $16.2 \pm 1.3$ | $0.118 \pm 0.007$ | $41.4 \pm 2.3$ | $0.207 \pm 0.017$ | $28.2 \pm 2.1$ | $0.033 \pm 0.003$ |
| gemini-2.5-pro | $71.8 \pm 7.7$ | $0.023 \pm 0.002$ | $\mathbf{9.6 \pm 2.6}$ | $\mathbf{0.067 \pm 0.018}$ | $41.8 \pm 2.3$ | $0.228 \pm 0.009$ | $9.4 \pm 3.7$ | $0.009 \pm 0.004$ |
| gemini-2.0-flash | $\mathbf{54.4 \pm 2.7}$ | $\mathbf{0.017 \pm 0.001}$ | $14.0 \pm 2.7$ | $0.094 \pm 0.014$ | $\mathbf{39.8 \pm 1.5}$ | $\mathbf{0.192 \pm 0.013}$ | $28.0 \pm 5.5$ | $0.038 \pm 0.009$ |

is dominated by the adjacency-removal phase. Following the analysis of (Kalisch & Bühlmann, 2007), the worst-case number of CI tests grows exponentially with the size of the largest adjacency set. Under our notation, for each unordered variable pair the algorithm may evaluate up to $2^k$ conditioning subsets when the adjacency set has maximum size $k$. Since there are $\mathcal{O}(N^2)$ such pairs, the total number of CI tests is bounded by $\mathcal{O}(N^2 2^k)$. An equivalent classical expression (Kalisch & Bühlmann, 2007; Spirtes et al., 2018) rewrites this as $\mathcal{O}(N^{k+1})$ by using the identity $\sum_{\ell=0}^{k} \binom{k}{\ell} = 2^k$ and bounding the number of nonempty adjacency configurations accordingly. When $k$ grows with $N$—as in dense graphs where $k = \Theta(N)$—PC-Stable becomes exponential in the number of variables, consistent with empirical observations of its rapidly increasing runtime in high-dimensional settings (Ha et al., 2014).

## D    Detailed Query Complexity Analysis

**ChatPC**  , which replaces the conditional independence (CI) tests in PC-Stable with LLM-based CI judgments, issues one query for each ordered pair of variables and for each conditioning set considered during adjacency removal. Using our notation, there are $\mathcal{O}(N^2)$ ordered variable pairs, and for each pair the algorithm enumerates up to $2^k$ conditioning subsets when the adjacency set has size at most $k$, where $k$ denotes the maximum adjacency size encountered during the search. Thus, the worst-case query complexity of ChatPC is $\mathcal{O}(N^2 2^k)$, which can be expressed as $\mathcal{O}(N^{k+1})$. This reduces to a polynomial bound when $k$ is small (sparse graphs) but becomes exponential in $N$ when $k = \Theta(N)$ (dense graphs).

**PairwisePrompt**   evaluates all $N(N-1)$ possible ordered edges using a single LLM query per pair, yielding a query complexity of $\mathcal{O}(N^2)$. BFSPrompt reduces this cost by exploring the graph in a breadth-first manner and issuing exactly one query per node expansion, giving a total of $\mathcal{O}(N)$ queries.

**Scalability and PromptBN**   requires only a small number of LLM invocations that do not scale with $N$. Scalability proceeds in three stages: (1) a facilitator LLM generates $M$ expert personas using one query; (2) each expert receives two prompts to propose and summarize candidate causal relations, contributing $2M$ additional queries; and (3) optional decycling prompts are issued only when an expert outputs a directed cycle. In the theoretical worst case, resolving all possible conflicts would require $\mathcal{O}(N^2)$ additional queries,

since there are at most $N(N-1)$ potentially inconsistent directed pairs. Hence, the algorithm's worst-case query complexity is $\mathcal{O}(N^2)$; however, with fixed $M$ and empirically rare cycles, the effective query cost behaves as $\mathcal{O}(1)$ with respect to $N$.

**PairwisePrompt+Data and BFSPrompt+Data** preserve the same interaction structure as in PairwisePrompt and BFSPrompt, and additionally incorporate observations into their prompts. Consequently, the query complexities of remain $\mathcal{O}(N^2)$ and $\mathcal{O}(N)$, respectively.

**LLM-CD** introduces a three-stage prompting strategy. The metadata derivation stage calls one query per variable, giving a cost of $\mathcal{O}(N)$. The causal extraction stage invokes the LLM once, using all enriched metadata, to extract the potential causal relationships. The final causal validation stage checks each extracted directed pair individually; in the worst case, the extraction stage may propose all $N(N-1)$ ordered pairs, resulting in $\mathcal{O}(N^2)$ validation queries. Thus, the overall worst-case query complexity of LLM-CD is $\mathcal{O}(N^2)$, dominated by the validation stage.

**ReActBN** alternates between LLM-guided refinement decisions and classical score evaluations. The number of LLM queries equals the number of refinement iterations prior to convergence. Although we denote the iteration count by $i$, in practice we fix it across datasets, mirroring the common setup in Hill Climbing. As a result, the practical query complexity of ReActBN behaves as $\mathcal{O}(1)$, making it substantially more query-efficient than existing hybrid methods such as LLM-CD, which require validation of all candidate edges. This allows ReActBN to leverage observational data for refinement while avoiding the large query overheads characteristic of prior LLM+Data techniques.

