# OpenReview forum: "Bayesian Network Structure Discovery Using Large Language Models"
_TMLR — Accepted by TMLR_

### Review · Reviewer_5trM · 2025-11-12

**Summary Of Contributions:**

1. A framework for Bayesian structure learning is proposed (PromptBN and ReActBN) that leverages Large Language Models to efficiently discover Bayesian Networks.

2. The iterative procedure of ReActBN, where an LLM agent can modify the graph based on scores computed from data, is a novel idea.

3. The experimental results presented in the paper are promising. However, the extent of the evaluation is quite limited.

**Additional Comments:**

**Questions**

1. **Variable Descriptions:** Were all variable descriptions human-made? As the method relies on an LLM, have the authors considered querying the LLM to *generate* these descriptions as well, and how might that impact performance?
2. **Source of Variance:** The standard deviations reported in the appendix are noted. Is this variance caused *only* by the LLM's stochasticity, or are there other sources of randomness in the experiments?
3. **Score Computation and Alternatives:** Could the authors provide a  more detailed breakdown of how the score in ReActBN is computed (is it always BIC)? Could other scoring functions or even more different approaches be used as well?
4. **Invalid Runs:** The paper mentions some runs were "invalid." Could the authors please clarify how often invalid runs occured and the specific reason for their invalidity? For instance, did they output cyclic graphs or fail to parse?
5. **Choice of LLMs:** What was the rationale for using different LLMs for PromptBN (GPT-4) and ReActBN (GPT-3.5)?
6. **ChatPC Complexity:** The $O(N^3)$ query complexity for ChatPC seems to assume only one query per conditioning set. Given that conditioning sets themselves grow worse than linearly, could the authors clarify how the $O(N^3)$ complexity is derived?
7. **LLM-CD Complexity:** Similarly, could the authors please provide a brief explanation or derivation for how the $O(N^2)$ worst-case complexity for LLM-CD is determined?
8. **Reasoning Model Performance:** The paper notes that reasoning-focused models performed worse. Do the authors have a hypothesis as to why this might be, or is this attributed primarily to experimental noise?

**Minor Suggestions**

- **"Ablation Study" Naming:** Please consider renaming the "Ablation Study." These experiments are valuable, but they don't seem to fit the standard definition of an ablation (i.e., removing a method component).
- **Table 7 Highlighting:** Please add boldface highlighting to Table 7 to indicate the best-performing methods, consistent with the other tables.
- **Figure 1 Font Size:** The text in Figure 1 is quite small. Please increase the font size for readability.
- **Citation Formatting:** There are numerous `citep`/`cite` formatting errors (e.g., 4.1 "Zhang et al. (2023)" and 5.1 "Following (Khatibi et al., 2024),"). Please perform a thorough check of all citations.
- **Section 5 Title:** The section title "Experiment" (singular) does not sound great. Please consider changing it to "Experiments" or "Experimental Evaluation."
- **Section Order:** The last sentence before 5.1. previews the sections out of order (mentions 5.6 before 5.5). Please correct this to match the paper's structure.
- **Table 3 Scaling:** The "scaled by x10^-3" in Table 3 is slightly confusing, as the text uses the decimal values. Please consider putting the full decimal values (e.g., 0.068) in the table for clarity and consistency.
- **Text/Table Mismatch:** The HC value for the CHILD dataset in the text (0.065) does not match the value in Table 3 (0.068). Please check and correct this.
- **Algorithm Line Numbers:** The line numbering in Algorithms 1 and 2 appears to be broken or rendered incorrectly.
- **Typo:** There is a typo in the text "250 ̆1000" (the " ̆" character).

**Audience:**

Yes

**Audience Explanation:**

Causal discovery is a problem many people are interested in. Apart from purely data-based and purely LLM-based methods, hybrid methods that aim to combine the strengths of both categories have begun to emerge. This paper proposes one such method that is novel and should be of some interest to the causal discovery community.

**Claims And Evidence:**

No

**Claims Explanation:**

1. **Incomplete Comparative Analysis:** Table 3 would be a good basis for experimental evaluation, but the overwhelming lack of results greatly weakens the validity of the results. For example, the purely LLM-based comparisons only show results on 2 or 3 of the 6 datasets, and many results are missing for the other methods as well. As I understand, the authors did not run these experiments themselves and only reported what was done in the original papers, correct? While this is generally not a problem, it leads to too many empty fields here, and I recommend running more of these experiments yourself.

2. **Evaluation on Lesser-Known Datasets:** I really like that the paper includes three lesser-known datasets (blockchain, covid, disputed3), especially since evaluating LLMs on famous datasets might always be biased, as they were likely to appear in their training data. It would be great if all experiments conducted for Table 3 could also be run on these datasets.

3. **Scalability:** A more extensive discussion on the performance of PromtBN and ReActBN on larger datasets and how well they scale would be helpful. For example, sorting the datasets by their variable size could also help illustrate how well the framework scales to larger problems.

**Requested Changes:**

1. **Experimental Results.** This is crucial for readers to assess the potential of this framework. Providing the missing results in Table 3 would greatly strengthen the experimental evaluation. Additionally, including the three less well-known datasets there (blockchain, covid, disputed3) would nicely complement these results.
2. **Questions.** Please respond to my questions listed in "Additional Comments" and consider clarifying these points in the manuscript as well. These should not be major points, but I think they could lead to changes that can improve the paper.

---

> ### Author Response · Authors · 2025-12-08
>
> We sincerely thank the reviewer for your prestigious time and valuable feedback! We are encouraged by the positive recognitions, including “The iterative procedure of ReActBN is a novel idea” and “really like that the paper includes three lesser-known datasets”. We will add multiple comments to address different concerns and questions from reviewers accordingly for better readability.

---

> > ### Author Response · Authors · 2025-12-08
> > **Reproduction Expansion: Unified Evaluation Across Classical and Lesser-Known Datasets**
> >
> > We thank the reviewer for raising both concerns regarding incomplete comparative analysis and the evaluation coverage on lesser-known datasets. We agree that strengthening the uniformity and breadth of baseline comparisons is essential for a meaningful assessment of our methods.
> >
> > **We have substantially expanded our experimental reproduction efforts. We rigorously audited all baselines and reproduced every method for which the available algorithmic descriptions and implementation details were sufficiently specified.** These methods were applied using an identical experimental setup **across all six classical bnlearn networks and all three newer networks, thereby eliminating gaps that previously arose from inconsistent reporting in the original papers**. This expansion directly addresses the reviewer’s suggestion to run more of these experiments ourselves and to include the newer datasets in all evaluations.
> >
> > A small number of prior methods, such as Scalability, could not be reliably reproduced due to incomplete or ambiguous specifications. We retained their published results solely for completeness and clearly marked them as imported. All other entries in the revised manuscript are newly reproduced under matched conditions, ensuring transparency and consistency. To provide a more coherent comparison, the revised manuscript consolidates all reproducible baselines—across both classical and newer datasets—into a single unified results table (revised Table 1), replacing the previous split across original Tables 3 (classical) and 4 (newer). This unified presentation emphasizes generalization performance and enables clear head-to-head comparisons across settings.
> >
> > The expanded results reinforce the validity of our conclusions. Under the 0-sample condition, PromptBN consistently exhibits strong recovery across nearly all datasets, including newer networks that are less likely to appear in LLM pre-training corpora. Under the 100-sample condition, ReActBN attains competitive or superior SHD and NHD scores on both classical and newer datasets. These trends persist under standardized and fully reproducible experimental conditions, indicating that the observed gains reflect genuine strengths of our methods rather than artifacts of missing entries or mixed reporting.
> > We appreciate the reviewer’s comments, which prompted a more comprehensive and transparent comparison. We believe the expanded reproducibility study and unified evaluation meaningfully strengthen the manuscript's empirical rigor.

---

> > ### Author Response · Authors · 2025-12-08
> > **Complexity Analysis and Scalability Discussion**
> >
> > We thank the reviewer for highlighting the importance of discussing how PromptBN and ReActBN scale with increasing graph size. In the revised manuscript, Section 6 now provides a comprehensive scalability analysis that examines both **CPU-side runtime complexity** and **LLM query complexity** across all methods. This extended discussion explicitly clarifies that **PromptBN requires only a single LLM invocation** ( $\mathcal{O}(1)$ query complexity), and that **ReActBN performs** $\mathcal{O}(N^2)$ **offline neighborhood enumeration per iteration while maintaining a practically constant query complexity** $\mathcal{O}(1)$.
> >
> > ## Scalability of PromptBN
> >
> > To further quantify PromptBN’s scalability, we analyzed its token usage using the same tokenizer as the o3 model. The results are shown below:
> >
> > | Dataset    | Variables | Total Tokens | Tokens per Variable | Metadata % |
> > |------------|-----------|--------------|---------------------|------------|
> > | cancer     | 5         | 618          | 32.0                | 25.9%      |
> > | asia       | 8         | 702          | 30.5                | 34.8%      |
> > | child      | 20        | 1,098        | 32.0                | 58.3%      |
> > | insurance  | 27        | 1,322        | 32.0                | 65.4%      |
> > | alarm      | 37        | 1,642        | 32.0                | 72.1%      |
> > | hailfinder | 56        | 2,250        | 32.0                | 79.6%      |
> >
> > As expected, the token count increases proportionally with the number of variables. Even for the largest dataset we evaluated (Hailfinder), the total prompt length is only around **2k tokens**. Since modern LLMs—led by models such as Gemini 3 Pro Preview (High) with a **2M-token context window**—typically support **128k–2M tokens**, PromptBN remains well within feasible context limits. Thus, **PromptBN does not present a practical scalability concern**.
> >
> > ## Scalability of ReActBN
> >
> > ReActBN uses a fixed prompt template and injects only the **top-k scores** ($k=10$ in our experiments) at each reasoning step. Therefore, **the prompt length remains effectively constant**, and scalability with respect to LLM query size is not an issue.
> >
> > Potential concerns instead arise on the CPU side. Here, ReActBN relies on the standard BIC score, the same portable scoring function widely used in classical structure-learning methods such as HC and other score-based algorithms. **ReActBN introduces no additional computational overhead** beyond the neighborhood enumeration that these methods already perform. Consequently, as the dataset size increases, **ReActBN remains feasible whenever Hill-Climbing (HC) itself is feasible**.
> >
> > We hope this expanded and clarified discussion fully addresses the reviewer’s concerns regarding the scalability of our methods.

---

> > ### Author Response · Authors · 2025-12-08
> > **Additional Comments 1-4**
> >
> > We appreciate the reviewer providing so many constructive and detailed comments to help us improve the quality of this manuscript. We have made the modification accordingly:
> >
> > **Variable Descriptions**
> > All variable descriptions used in our experiments are human-written. For the classical benchmarks, we used the official descriptions provided in the bnlearn documentation; for the unseen datasets, we adopted curated descriptions from BnRep (https://manueleleonelli.shinyapps.io/bnRep/) and dataset pages indexed at https://r-packages.io/datasets. None were generated by an LLM.
> >
> > We considered using an LLM to generate descriptions, but this approach is unreliable for domain-specific datasets with technical abbreviations (e.g., Hailfinder), where generated descriptions often become vague or incorrect. In realistic applications, practitioners typically know the semantic meaning of variables even when data are limited, consistent with our data-free setting. Therefore, using accurate human-written descriptions yields a more faithful evaluation of the structural-discovery component.
> >
> > **Source of Variance**
> > Yes, the only randomness comes from the LLM.
> >
> > **Score Computation and Alternatives**
> > In our reported experiments, ReActBN uses the Bayesian Information Criterion (BIC), matching the scoring used by the HC baseline to ensure fair comparison. The framework itself, however, is score-agnostic: the scoring component is fully modular, and any alternative—such as BDeu or other likelihood- or information-theoretic criteria—can be substituted without modifying the algorithm. We have verified in local tests that BDeu integrates seamlessly with ReActBN.
> >
> > At each iteration, ReActBN retrieves scores computed offline using pgmpy and incorporates these numerical values into the prompt so the LLM can reason about the next action. We will release the codebase with a modular scoring interface to support additional or user-defined scoring functions.
> >
> >
> > **Invalid Runs and the Definition of Validity**
> > We appreciate the reviewer’s request for clarification. In our evaluation protocol, a run is marked invalid if the LLM (i) fails to generate an output in the required JSON format or (ii) produces a graph that does not pass our dual validation procedure, which checks parsing correctness, acyclicity, and structural consistency. To ensure comparability across methods, each configuration must complete five valid runs. When an invalid run occurs, our error-handling mechanism automatically re-initiates the run until a valid output is obtained; if retries reach a predefined limit, the configuration is conservatively recorded as a failure.
> >
> > We also believe the reviewer may be referring specifically to the “Unparsable” errors observed in our model-selection experiments. These entries denote cases where the response could not be parsed into a JSON object and therefore failed before entering the dual validation pipeline. This issue occurred only with deepseek-v3, and it was exceedingly rare. Because no other model exhibited comparable failures and the few occurrences did not follow a consistent pattern, we do not have a meaningful statistical distribution to report. Nonetheless, all such cases were systematically excluded through our validation and retry mechanism to ensure the reliability of the final results.

---

> > ### Author Response · Authors · 2025-12-08
> > **Additional Comments - Choice of LLMs**
> >
> > Thank you for raising this point regarding model selection. PromptBN is evaluated using o3-2025-04-16, and ReActBN is implemented with gpt-4.1-2025-04-14. Our choices reflect the differing roles the LLMs play in the two stages of our framework.
> >
> > The o-series models are optimized for structured multi-step reasoning, problem decomposition, and consistent schema generation. PromptBN requires the model to internally interpret metadata, infer plausible causal relationships, and construct a complete, coherent DAG in one shot. These demands align well with the strengths of the o-series. Additionally, o3 has several readily available variants (o3-mini, o3, o3-pro), which enables our later model selection study examining how model capacity affects PromptBN’s performance. At the time of experimentation, the o1 family had already been deprecated, and o4-mini was the only released o4 variant, making o3 the most stable and suitable choice.
> >
> > In contrast, the refinement phase in ReActBN is guided by external scoring signals (e.g., BIC) and an explicit ReAct reasoning scaffold. Here, the LLM’s primary responsibility is not to perform deep internal causal reasoning, but to produce stable, instruction-following, step-by-step actions that integrate cleanly with the algorithmic loop. The gpt-4.1 model is designed as a balanced, reliable general model with strong adherence to instructions and low variance in stepwise reasoning, which makes it a better fit for this iterative decision-making setting. It also offers a favorable cost–stability tradeoff for repeated calls.
> >
> > In summary, the two models were chosen intentionally to align with the distinct cognitive demands of each stage: **o3 for single-shot structured reasoning**, and **gpt-4.1 for stable iterative refinement**. We have clarified this rationale in the updated manuscript.

---

> > ### Author Response · Authors · 2025-12-08
> > **Additional Comments - Reasoning Model Performance**
> >
> > We currently refrain from offering a definitive hypothesis, as this trend has not been consistently reported in prior work, and our present results alone are insufficient to attribute it to a specific cause. Our intention was to transparently report the empirical observation rather than infer mechanisms without adequate evidence. While potential factors such as architectural design or prompt sensitivity may play a role, isolating these effects requires additional controlled ablations that were beyond the scope of this submission. We view this as an important direction for future investigation.

---

> > ### Author Response · Authors · 2025-12-08
> > **Additional Comments - ChatPC Complexity and LLM-CD Complexity**
> >
> > As mentioned in the complexity analysis section, a more complete complexity analysis is now included in the revised manuscript. For clarity, the finalized complexities for ChatPC and LLM-CD(HC) are summarized below:
> >
> > | **Algorithm**   | **Category** | **Time Complexity**            | **Query Complexity**                         |
> > |-----------------|--------------|--------------------------------|-----------------------------------------------|
> > | ChatPC          | LLM          | $\mathcal{O}(N^2 \cdot 2^k)$ | –                                             |
> > | LLM-CD(HC)      | Hybrid       | $\mathcal{O}(N^2)$           | $\mathcal{O}(1)^* \text{ to } \mathcal{O}(N^2)$ |
> >
> > **ChatPC**
> > In ChatPC, each conditional independence (CI) test is replaced by a single LLM query. Thus, the overall complexity depends on:
> > * the number of unordered variable pairs, which is $\mathcal{O}(N^2)$;
> > * the number of conditioning subsets examined for each pair.
> > Following the standard PC-Stable analysis, the number of possible conditioning subsets is exponential in the maximum adjacency size. In particular, if $k$ denotes the largest adjacency set encountered during edge-removal, the number of CI tests per pair is $2^{k}$. Therefore, the total complexity becomes $\mathcal{O}(N^2 \cdot 2^k)$. We clarified this term in the revision because the earlier version misstated the complexity of the conditioning-subset enumeration. The corrected form above matches the classical analysis of PC-Stable and provides a precise characterization of ChatPC’s computational behavior.
> >
> > **LLM-CD**
> > The query complexity of LLM-CD follows directly from its three-stage prompting design. The metadata-derivation stage requires one query per variable, yielding $\mathcal{O}(N)$. The causal-extraction stage uses a single query to propose candidate directed pairs. The complexity is dominated by the final validation stage, where each proposed edge is checked individually. In the worst case, the extraction stage may output all $N(N-1)$ ordered pairs, resulting in $\mathcal{O}(N^{2})$ validation queries. Therefore, the overall worst-case query complexity of LLM-CD is $\mathcal{O}(N^{2})$.

---

> > ### Author Response · Authors · 2025-12-08
> > **All minor improvement suggestions**
> >
> > 1. **"Ablation Study" Naming**: We have now restructured it as a separate section named “Factor Analysis”, where we believe the sample size or the models used are the factors that may impact the final results.
> > 2. **Table content**: We re-checked all the table entries to ensure we used the right number in the content. We streamlined the tables by using the absolute value of NHD instead of the confusing “x10^-3” scaling. We use 1-digit decimal for all SHD and 3-digit decimal for NHD, and highlight the best performance in bold font across all tables.
> > 3. **Citation Formatting**: We have re-checked all citations and corrected them accordingly.
> > 4. **Figure 1 Font Size**: We will work on this in the next revision and ensure its readability.
> > 5. **Section 5 Enhancement**: We have updated the title to ‘Experiments’, corrected the out-of-order section preview before Section 5.1, and restructured the subsections for better clarity and flow.
> > 6. **Algorithm Line Numbers**: We have fixed this.
> > 7. **Typo in 250~1000**: We have removed this as we reproduced the algorithms in our identical experimental setup (sample size=100).

---

> ### Comment · Reviewer_5trM · 2025-12-11
>
> Thank you for the detailed revision! I think the experimental results now sufficiently show the value of the proposed approaches. In addition, the responses to my and the other reviewers' comments clarified other points, and the changes made to the manuscript incorporate these clarifications well.
> I have only two minor comments remaining:
> 1. The manuscript states, "Across all data-available settings, our proposed ReActBN achieves the strongest performance on nearly all classical benchmarks. ReActBN exactly recovers the Cancer network (SHD = 0) and attains the best results on 7 out of 9 datasets". This doesn't seem to match the results shown in Table 1, where ReActBN attains the best results on 6 out of 9 datasets in these settings (LLM-CD(HC) for Asia, PromptBN+HC for Hailfinder, and GES for covid lower scores). The text should be changed to correct or clarify this. The results are still sufficiently strong, I only want to correct this small error.
> 2. There is a "." missing at the end of page 5.
> I have no major concerns remaining.

---

> > ### Author Response · Authors · 2025-12-17
> >
> > We sincerely thank the reviewer for the careful review, the time and effort invested, and the constructive feedback throughout the review process. We greatly appreciate the positive assessment of our revisions and the helpful minor comments, which will help us identify issues we missed. We will update the manuscript accordingly in the next revision. Thank you again for your valuable insights and support!

---

### Review · Reviewer_zSLz · 2025-11-19

**Summary Of Contributions:**

**Summary.**

The paper proposes a two‑phase, LLM‑centric framework for Bayesian Network (BN) structure discovery:

- **Phase 1 (PromptBN):** single‑call meta‑prompting that elicits a full DAG from variable metadata, with dual output formats (parents‑per‑node and edge list) and acyclicity/consistency checks; regeneration on failure.
- **Phase 2 (ReActBN):** an agentic refinement loop that exhaustively enumerates local neighbors (add/remove/flip), scores them with BIC, surfaces top‑k candidates, and lets the LLM pick an action (or terminate) under a tabu regime.

Experiments on 7 classic bnlearn networks and 3 bnRep networks compare PromptBN/ReActBN with existing LLM-based approaches and classical methods (HC, PC-Stable, GES, rIeB, rBFchi2) using SHD/NHD, and discuss LLM query complexity. Ablations vary in sample size and LLM family.

**Strengths.**

S1: The PromptBN / ReActBN framework is well-structured and clearly presented, with both high-level intuition (Figure 1) and pseudocode (Algorithms 1 and 2). This makes it relatively easy for readers to understand how the system works end-to-end.

S2: On several benchmark networks, PromptBN alone recovers reasonable structures, and using PromptBN as an initializer or ReActBN as a refinement step often improves over plain HC at 100 samples, especially on Child, Insurance, Alarm, and two of the three newer bnRep networks.

S3: Ablation studies (sample size and LLM choice) are helpful and generally consistent with intuition.

**Weaknesses.**

W1: Algorithm 2 accepts *any* LLM-selected move from the top-k candidates, even when the score difference Δ is negative. There is no explicit constraint “only accept if score improves” and no tracking of the best-scoring graph seen so far; the algorithm simply returns the last graph visited.

W2:  There is no score‑only top‑k (no‑LLM) or top‑k random control using the same enumeration/BIC/tabu for ReActBN, so the incremental value of the LLM’s choice in Phase 2 is unclear.

W3: NHD is defined using a denominator of N² while described as “normalized by the total number of possible edges.” For directed graphs without self-loops, this is N(N−1); clarifying that they normalize by N² adjacency entries (including diagonal) or adjusting the description would improve clarity.

W4: The BIC scoring setup is not fully specified (e.g., discrete vs continuous variables, parameter estimation method, implementation details), which could matter for reproducibility and for interpreting SHD/NHD differences.

W5: While the prompt structure is described, specific details required for reproduction are missing or risky. The paper uses specific model snapshots (e.g., `o3-2025-04-16`), but code is only promised "upon publication." Given the sensitivity of LLM-based evaluations to prompt wording and temperature settings, this hinders reproducibility.

**Additional Comments:**

This is a timely and clearly written paper with an appealing high-level idea: use LLMs both to propose an initial BN from metadata and to guide a score-based refinement loop. The experiments are reasonably extensive, and the ablations are useful.

However, I have substantial concerns about:

- The clarity and correctness of the ReActBN search procedure as currently specified.
- The lack of a score-only / random control to isolate the LLM’s contribution in ReActBN.
- The limited analysis of computational cost beyond LLM query counts.
- The potential for pre-training overlap in “data-free” experiments.
- The mixing of in-house vs imported baselines with different conditions.

**Audience:**

Yes

**Audience Explanation:**

The paper sits at the intersection of Bayesian networks, causal/structure discovery, and LLM-based reasoning/agents, all topics of active interest at TMLR. A method that can generate BNs from metadata alone and refine them with a ReAct-style loop is relevant to readers working on hybrid neuro-symbolic systems, LLM-guided search, and data-efficient structure learning.

**Broader Impact Concerns:**

I do not see major ethical red flags requiring a long Broader Impact section, but one point is worth emphasizing: the paper promotes LLM-assisted discovery of probabilistic/causal structures. If deployed in high-stakes domains (e.g., medicine, policy, finance) without strong domain-expert validation, there is a risk of over-trusting partially-validated structures. A short note reminding practitioners that these BNs should be treated as hypotheses rather than ground truth would be helpful.

**Claims And Evidence:**

No

**Claims Explanation:**

C1: The efficiency discussion and Table 5 focus on LLM query complexity (O(1) for PromptBN and ReActBN) but do not analyze the computational cost of repeated neighbor enumeration and BIC rescoring, which is O(N²) neighbors per iteration times the number of iterations. In practice, this could become expensive for larger graphs, even if the number of LLM calls is small. Yet the paper presents the method as efficient without discussing CPU runtime.

C2: Most classic BN benchmarks used here are textbook networks that LLMs might have seen during pre-training; the “data-free” story, therefore, needs a more explicit discussion of pre-training overlap vs. genuine reasoning.

C3: Table 3 mixes numbers reproduced here (PC‑Stable/HC at 100 samples with pgmpy *defaults*) with results imported from other papers at unknown or different sample sizes (e.g., GES; several LLM+Data entries). This weakens head‑to‑head conclusions and may overstate gains.

**Requested Changes:**

Solve or clarify W1-5, C1-3.

---

> ### Author Response · Authors · 2025-12-08
>
> We sincerely thank the reviewer for your prestigious time and valuable feedback! We are encouraged by our recognized strengths, such as “a timely and clearly written paper with an appealing high-level idea” and “the ablations are useful”. We will add multiple comments for different aspects to address the reviewer's concerns and questions.

---

> > ### Author Response · Authors · 2025-12-08
> > **LLM’contribution in ReActBN**
> >
> > We thank the reviewer for raising this important point. We have now added both required controls and clarified their relationship to ReActBN. First, **PromptBN+HC** serves as the correct score-only baseline, as it runs Hill Climbing directly on the PromptBN-initialized graph and relies solely on BIC scores to greedily select the next best action. Second, following the reviewer’s suggestion, **we implemented a top-k random control (PromptBN+RandomChoice), which samples uniformly from the same observation set used by ReActBN—namely, the top-10 legal operations plus the termination option**—while using the same initial graphs from PromptBN. Together, these two variants provide a complete set of comparisons that isolate the contribution of LLM in ReActBN.
> >
> > Additionally, we clarify that we do not apply tabu search within ReActBN. Because ReActBN allows the LLM to select actions with a negative score delta, it may choose to revisit previously explored graphs. To prevent such cycles while keeping the algorithm faithful to the ReAct-style decision process, we maintain a tabu list solely to avoid re-visiting previously encountered graphs.
> >
> > | Algorithm                | Asia SHD | Asia NHD | Cancer SHD | Cancer NHD | Child SHD | Child NHD | Insurance SHD | Insurance NHD | Alarm SHD | Alarm NHD | Hailfinder SHD | Hailfinder NHD |
> > |--------------------------|----------|----------|------------|------------|-----------|-----------|----------------|----------------|-----------|-----------|------------------|------------------|
> > | PromptBN+HC              | 8.0      | 0.094    | 0.0        | 0.000      | 18.2      | 0.030     | 46.6          | 0.055          | 43.6      | 0.028     | 63.8             | 0.018           |
> > | PromptBN+RandomChoice    | 5.6      | 0.084    | 1.2        | 0.056      | 23.4      | 0.053     | 43.4          | 0.056          | 43.0      | 0.031     | 79.2             | 0.026           |
> > | ReActBN (Ours)           | 6.4      | 0.075    | 0.0        | 0.000      | 18.0      | 0.028     | 40.2          | 0.049          | 35.4      | 0.024     | 75.0             | 0.023           |
> >
> > The results, updated in Table 1(main result, previously Table 3), show that **ReActBN outperforms both the score-only baseline and the random control on 5 out of 6 datasets across both SHD and NHD metrics**. These findings demonstrate that ReActBN’s improvements are not attributable to greedy scoring or randomness; rather, the LLM meaningfully integrates the score observations and performs a reasoning-driven action selection that yields consistently stronger refinements. We therefore believe that **W2** has been fully addressed.

---

> > ### Author Response · Authors · 2025-12-08
> > **Computational Cost/Monetary Analysis and Scalability**
> >
> > We thank the reviewer for the insightful comments regarding computational efficiency and scalability in **C1**. Our initial discussion focused on highlighting the LLM query efficiency of our methods, as query cost is often the dominant limiting factor in LLM-based algorithms. We appreciate the reviewer’s observation that a complete assessment should also articulate the CPU-side cost of neighborhood enumeration and BIC rescoring. In response, we have added a thorough analysis in Section 6 that provides a rigorous treatment of both CPU-side runtime complexity and LLM query complexity, followed by a discussion of their implications for scalability across all methods. A consolidated comparison table summarizing the complexities of PromptBN, ReActBN, and all prior baselines has also been included.
> >
> > A summarized complexity analysis table can be found below:
> > | **Algorithm**         | **Category** | **Time Complexity**                 | **Query Complexity**              |
> > |-----------------------|--------------|-------------------------------------|-----------------------------------|
> > | Scalability           | LLM          | –                                   | $$\mathcal{O}(1)^* \text{ to } \mathcal{O}(N^2)$$ |
> > | PairwisePrompt        | LLM          | –                                   | $$\mathcal{O}(N^2)$$              |
> > | BFSPrompt             | LLM          | –                                   | $$\mathcal{O}(N)$$                |
> > | ChatPC                | LLM          | $$\mathcal{O}(N^2 \cdot 2^k)$$      | –                                 |
> > | **PromptBN (Ours)**   | LLM          | –                                   | $$\mathcal{O}(1)$$                |
> > | PC-Stable             | Data         | $$\mathcal{O}(N^2 \cdot 2^k)$$      | –                                 |
> > | HC                    | Data         | $$\mathcal{O}(N^2)$$                | –                                 |
> > | GES                   | Data         | $$\mathcal{O}(N^2)^* \text{ to } \mathcal{O}(N^3)$$ | –                   |
> > | LLM-CD(HC)            | Hybrid       | $$\mathcal{O}(N^2)$$                | $$\mathcal{O}(1)^* \text{ to } \mathcal{O}(N^2)$$ |
> > | **ReActBN (Ours)**    | Hybrid       | $$\mathcal{O}(N^2)$$                | $$\mathcal{O}(1)$$                |
> >
> > \* the value represents the best case depending on an internal step’s LLM generation.
> >
> >
> >
> >
> > First, we now explicitly analyze the offline runtime of ReActBN. Each refinement step enumerates $\mathcal{O}(N^{2})$ legal moves, each requiring a BIC calculation. Although the theoretical cost is $\mathcal{O}(T \cdot N^{2})$, the iteration budget $T$ is fixed to a small constant in practice. Thus, the effective complexity remains $\mathcal{O}(N^{2})$, matching classical methods such as HC and hybrid approaches like LLM-CD. This shows that the offline computations do not form a bottleneck for ReActBN.
> >
> > Second, we emphasize that query complexity is often the dominant bottleneck for LLM-only or hybrid methods. While sharing the same offline runtime as other baselines, ReActBN guarantees $\mathcal{O}(1)$ query complexity, providing a clear efficiency advantage over LLM-CD.
> >
> > Finally, we extend our discussion on scalability.
> > * PromptBN’s limitation arises from requiring all variable metadata in a single prompt, but modern LLMs, which usually have a context window size of 128k or above, can typically accommodate the sizes of standard BN benchmarks – the largest dataset Hailfinder in our experiment has 2.2k tokens in the prompt.
> > * ReActBN adds only a constant number of scores to the prompt at each reason step, keeping prompt length stable as the graph grows. Combined with its $\mathcal{O}(N^{2})$ offline cost—no worse than HC or LLM-CD—ReActBN scales to any dataset on which HC itself remains feasible.

---

> > ### Author Response · Authors · 2025-12-08
> > **Contamination Tests: Addressing Pre-Training Overlap and Genuine Reasoning**
> >
> > We appreciate the reviewer’s concern in C2 regarding potential overlap between classic bnlearn networks and LLM pre-training corpora. Because modern LLM training data are undisclosed, such overlap cannot be ruled out. To directly assess whether PromptBN’s performance may stem from memorization, we conducted three controlled contamination settings: (1) Variable-Name Only, (2) Variable-Description Only, and (3) Scrambled Order, where the full metadata is preserved but randomly permuted.
> >
> >
> > | Constraints            | Asia SHD↓ | Asia NHD↓ | Cancer SHD↓ | Cancer NHD↓ | Child SHD↓ | Child NHD↓ | Insurance SHD↓ | Insurance NHD↓ | Alarm SHD↓ | Alarm NHD↓ | Hailfinder SHD↓ | Hailfinder NHD↓ |
> > |----------------------|-----------|-----------|--------------|--------------|------------|-------------|------------------|------------------|-------------|-------------|-------------------|-------------------|
> > | VarName Only         | 0.0       | 0.000     | 0.67         | 0.030        | 27.3      | 0.050       | 41.3            | 0.050            | 43.8       | 0.030       | 118.67            | 0.040            |
> > | VarDescription Only  | 0.0       | 0.000     | 0.33         | 0.010        | 31.3      | 0.070       | 44.3            | 0.060            | 43.8       | 0.030       | 119.83            | 0.040            |
> > | ScrambleOrder        | 0.0       | 0.000     | 0.6          | 0.024        | 24.8       | 0.051       | 37.4             | 0.044            | 48.4        | 0.964       | 97.8              | 0.032            |
> >
> > The results have been added to Table 1 (previously Table 3 and 4) in the revised paper. We observe that for larger classic benchmarks such as Child, Insurance, Alarm, and Hailfinder, the model fails to recover the graphs well in both VarName-only and VarDescription-only settings, even when the original variable names or descriptions are available. In contrast, small networks such as Asia and Cancer remain almost perfectly recoverable under all variants, which we interpret as consistent with possible training-data contamination on these very well-known toy examples. This pattern suggests that simple memorization of names or descriptions alone is insufficient to explain our results on larger networks, and supports our claim that PromptBN leverages genuine reasoning from metadata.
> >
> > We note a modest decline on Hailfinder in the permutation condition, likely due to the well-known lost-in-the-middle effect associated with long contexts. Importantly, this does not alter the overall conclusion: across tests, the contamination analysis consistently shows that PromptBN’s strong performance arises from genuine reasoning over metadata rather than pre-training overlap.

---

> > ### Author Response · Authors · 2025-12-08
> > **Mixing of In-house vs Imported Baselines with Different Conditions**
> >
> > We appreciate the reviewer’s concern regarding the mixture of in-house reproduced baselines and results imported from prior work that did not report or align their sample sizes. This inconsistency stems from the fact that many earlier studies did not provide results for all datasets or for small-sample conditions, making it impossible to obtain a fully uniform set of baselines. These gaps are inherited from the literature rather than arising from our experimental design.  Therefore, we respectfully disagree that this mixture weakens our conclusions or overstates the gains. We acknowledge that this heterogeneity warrants clarification.
> >
> > To address this concern, **we rigorously audited all baselines and reproduced every method for which the available algorithmic description and implementation details were sufficient. These methods were evaluated under the identical experimental setup used in our study across all six classical and three newer networks, enabling consistent head-to-head comparisons for the majority of baselines**. We retained the published results for Scalability solely to preserve completeness and clearly marked them as imported, because it could not be reliably reproduced due to incomplete or ambiguous specifications.
> >
> > To further improve transparency, **the revised manuscript consolidates all reproducible baselines into a single unified table** (revised Table 1), replacing the earlier split between Tables 3 (classical datasets) and 4 (newer datasets). All entries are reproduced unless they are explicitly labeled as imported, ensuring that conclusions rely only on results obtained under matched experimental conditions.
> >
> > Finally, the consolidated results confirm that our conclusions are not overstated. Under the 0-sample setting, PromptBN delivers consistently strong performance across most datasets; under the 100-sample setting, ReActBN achieves competitive or superior SHD and NHD scores across both classical and newer benchmarks. These outcomes arise from standardized and reproducible evaluations, demonstrating genuine methodological advantages rather than artifacts of mixed reporting. We thank the reviewer for highlighting this point, which helped us strengthen the clarity and rigor of our comparisons. We believe these results reflect genuine methodological advantages rather than inflated comparisons.

---

> > ### Author Response · Authors · 2025-12-08
> > **Additional Comments**
> >
> > **Reproducibility**
> >
> > The OpenAI response API no longer accepts temperature as a parameter. Therefore, we did not set the temperature for our OpenAI models and set it to 0 by default for the non-OpenAI models. The codebase is now available at https://github.com/anonymous-4-review-only/my-anonymous-project-001. We have open-sourced exact prompting, source code of both methods, and our reproduction implementation of compared baselines.
> >
> >
> > **Definition and Description of NHD**
> >
> > We thank the reviewer for pointing out the inconsistency between the content and the formula definition in **W3**. We have updated the description accordingly.

---

> ### Author Response · Authors · 2025-12-08
> **Clarification on the ReActBN search procedure**
>
> ReActBN follows the ReAct (Reasoning + Acting) paradigm, where an LLM functions explicitly as an agent that alternates between offline procedural actions (acting component) and high-level reasoning (reasoning component), which incorporates the observation from the last action. In our setting, the acting component executes all LLM-off operations—the scoring methods, identical to classical algorithms. We select the top-k scored neighbors as the observations. The reasoning component is then performed purely by the LLM. Observations received and integrated with the reasoning guideline, LLM then decides the next action to take. Instead of following a defined rule, the agent uses these observations and its internal knowledge to select a move (including termination).
>
> Regarding the reviewer’s concern in **W1** (no “accept only if improved” rule, no best-score tracking, and returning the last visited graph): this behavior is intentional and reflects the design of ReActBN as an agent-driven exploratory procedure rather than a deterministic optimization algorithm. Because the LLM is explicitly responsible for determining when the search should stop, the output is defined as the final graph selected by design, rather than the best-scoring graph along the path. Thus, there is no need to track the best graph. We will better clarify this design choice in the revised manuscript.
>
> **BIC Score Selection and Implementation**
>
> Since the scoring method is just the act that provides observations for LLM to better reason and decide, the BIC score is not the only option. We also allow other scoring method. In our experiments, we use the BIC score from the widely used Python package pgmpy by default in both HC and ReActBN. Regarding the reviewer’s concern in **W4**, according to the pgmpy documentation (https://pgmpy.org/structure_estimator/hill.html#pgmpy.estimators.BIC), it works on discrete networks, and the parameter estimation method is Maximum Likelihood Estimation. For completeness, we clarify that all variables in our experiments use the discrete-CPT parameterization assumed by pgmpy’s BIC implementation, and no discretization or additional preprocessing is applied.

---

> ### Comment · Reviewer_zSLz · 2025-12-10
>
> Thank you for the clarification! Most of my concerns have been resolved. Regarding W1, which remains my major concern: while I understand that ReActBN is designed as an exploratory agent in which the LLM determines termination, this design choice raises significant concerns about performance stability. Specifically, because the method allows selecting worse-scoring moves and strictly returns the last-visited graph rather than the best graph encountered, the final output is highly susceptible to the stochastic nature of the LLM’s decisions, with no fallback guarantee of quality. At present, I do not see sufficient evidence, from either theoretical or experimental perspectives, that this agentic decision-making reliably yields improvements over standard hill-climbing or even a controlled random walk.

---

> > ### Author Response · Authors · 2025-12-17
> > **Empirical Evidence for the Stability and Reliability of ReActBN: Additional Experiments**
> >
> > We directly evaluated whether ReActBN degenerates into a constrained random walk by introducing a **PromptBN + RandomChoice** baseline, which operates in the identical action space but selects actions uniformly at random. As reported in our earlier response, ReActBN outperforms this baseline on **5 of 6 benchmark datasets** under both Structural Hamming Distance (SHD) and Normalized Hamming Distance (NHD). Notably, despite strong structural priors induced by PromptBN, the performance of PromptBN + RandomChoice  remains substantially weaker. This provides direct empirical evidence that ReActBN’s gains arise from **state-aware, goal-directed agentic decision making**, rather than stochastic wandering.
> >
> >
> > To address concerns about stochasticity, termination, and the absence of an explicit best-so-far safeguard, we conducted additional controlled experiments (Section 7). We compare ReActBN against Hill Climbing (HC) under **best PromptBN**, **worst PromptBN**, and **empty (HC-only)** initializations, using **five independent runs each with randomly sampled 100 samples**. To take the stochasticity of LLMs into consideration, on each sample set, we repeat ReActBN, five times, yielding **25 runs**. We reporthe t mean and standard deviation (std) across runs.
> >
> > | Dataset        | HC (Best PromptBN Init) | ReActBN (Best PromptBN Init) | HC (Worst PromptBN Init) | ReActBN (Worst PromptBN Init) |
> > |----------------|-----------------|----------------------|-------------------------|------------------------------|
> > | Alarm        | 39.2 ± 4.44   | 37.6 ± 1.22     | 38.4 ± 4.10             | 39.6 ± 2.26                 |
> > | Insurance  | 47.2 ± 0.84   | 35.3 ± 0.99    | 48.2 ± 3.56             | 39.6 ± 2.02                 |
> > | Hailfinder   | 82.4 ± 2.70   | 79.5 ± 1.53    | 85.2 ± 2.49             | 83.5 ± 2.18                 |
> > | Child          | 28.2 ± 2.59    | 23.9 ± 2.69   | 27.0 ± 2.35             | 26.2 ± 2.38                 |
> > | blockchain  | 12.0 ± 1.00   | 13.4 ± 1.88    | 13.0 ± 1.87             | 12.9 ± 2.25                 |
> > | covid           | 32.4 ± 1.52   | 32.4 ± 1.53    | 31.6 ± 0.55             | 31.7 ± 0.59                 |
> > | disputed3   | 11.0 ± 4.74   | 10.2 ± 2.75     | 17.2 ± 4.97             | 17.5 ± 2.03                 |
> >
> > Under best PromptBN initialization, ReActBN exhibits lower **run-to-run variance** than HC on 3 of 7 datasets, with the remaining cases being comparable or slightly higher. Under the worst PromptBN initialization, ReActBN shows **lower variance on 4 of 7 datasets**, one comparable case, and two higher-variance cases. Representative examples include:
> > * Alarm (best init): ReActBN 37.6 ± 1.22 vs. HC 39.2 ± 4.44 (≈4x std reduction, ≈13x variance reduction).
> > * disputed3 (worst init): ReActBN 17.5 ± 2.03 vs. HC 17.2 ± 4.97 (>2x std reduction, >6× variance reduction).
> >
> > These results demonstrate that LLM-determined termination does **not lead to unstable outcomes** in practice. Despite returning the terminal state rather than the best-so-far solution, ReActBN consistently converges with low variance across datasets and initialization regimes.
> >
> >
> > More broadly, **stochasticity does not imply instability**. Empirically, ReActBN is often **more stable than HC**, particularly when initialization quality degrades and data is limited. HC is highly sensitive to poor starting points and strongly replies on data likelihood, whereas ReActBN maintains comparable or lower variance while achieving competitive or superior mean SHD.
> >
> >
> > Allowing non-monotonic moves and returning the terminal state are **standard design choices** in combinatorial optimization and Bayesian network structure learning (e.g., simulated annealing, tabu search, and MCMC-based methods). ReActBN follows this well-established paradigm. Crucially, across all evaluated settings, we observe **no catastrophic degradation**: ReActBN consistently matches or improves upon HC and substantially outperforms empty or random baselines.
> >
> >
> > Finally, while ReActBN does not offer formal optimality guarantees, this is consistent with classical score-based structure learning methods, including HC and its variants. In lieu of theory, our extensive empirical evaluation demonstrates that ReActBN delivers **stable, reliable, and competitive performance**, which is the primary criterion for practical Bayesian network structure learning.

---

> > > ### Comment · Reviewer_zSLz · 2025-12-22
> > >
> > > Thanks for the additional experiments; although the method still lacks principled guarantees, I appreciate the empirical evidence demonstrating improved stability and competitive performance, and I no longer have major concerns.

---

### Review · Reviewer_UvjX · 2025-11-23

**Summary Of Contributions:**

The paper proposes a two‑phase LLM‑centered framework for Bayesian network (BN) structure discovery from variable metadata and, optionally, observational data. Specifically, in the data-free case, the paper introduces PromptBN to query LLMs with metadata, and when observational data are available, the paper introduces ReActBN, which integrates the ReAct reasoning paradigm with structure scores such as the Bayesian Information Criterion (BIC) for iterative refinement.

Experiments cover seven classic bnlearn networks and three newer bnRep networks. PromptBN is compared to prior LLM‑based methods and traditional data‑driven structure learners. ReActBN is compared to these data‑driven baselines and to a hybrid PromptBN+HC initialization scheme. Experiments show that the proposed method achieves strong performance. In particular, it accurately recovers exact structures on small networks and maintains high accuracy on medium-sized graphs, even with limited data.

**Audience:**

Yes

**Audience Explanation:**

The paper clearly targets BN structure discovery from metadata, and makes an explicit distinction between data‑free and data‑aware settings. The two‑phase design (PromptBN + ReActBN) is conceptually simple and well‑motivated by limitations of both pure LLM elicitation and pure data‑driven search. The experiments are comprehensive and demonstrate the effectiveness of the proposed two methods.

**Broader Impact Concerns:**

No broader impact concerns.

**Claims And Evidence:**

No

**Claims Explanation:**

There are two major concerns:
1. Training-data contamination: Many classic BN benchmarks (Asia, Cancer, Alarm, Child, Insurance, Hailfinder, Earthquake) and their graphs have been widely used and documented for years; they are very likely present (at least partially) in the training corpora of modern LLMs. The strong performance of PromptBN in the “0‑sample” setting, especially on well‑known benchmarks (e.g., exact recovery on Asia and Cancer), could plausibly be explained by memorization rather than genuine reasoning from metadata.

2.  Complexity and scalability claims are somewhat overstated: The paper repeatedly claims O(1) query complexity for PromptBN and ReActBN. This is true when the number of LLM calls is a function of the number of nodes N, but: (1) ReActBN performs O(N²) offline work per iteration by enumerating all possible local moves for all ordered node pairs, computing scores, and selecting top‑k. (2) The number of iterations T is a hyperparameter. So total offline complexity is O(T·N²), which can be substantial for large graphs and is not analyzed empirically.

Thus, “O(1) query complexity” may be technically correct but somewhat misleading: wall‑clock cost scales significantly with N, and there is no runtime or token‑count analysis. A more nuanced treatment of both computational and monetary cost (tokens/API calls) would be desirable.

**Requested Changes:**

1. Provide a more explicit and nuanced discussion of the risk of data contamination to validity, possibly with simple tests such as: 1) masking or perturbing variable names and descriptions, 2) scrambling node labels and checking whether performance degrades.



2. Provide a more in-depth analysis of the complexity of the proposed methods. A more nuanced treatment of both computational and monetary cost (tokens/API calls) would be desirable.

---

> ### Author Response · Authors · 2025-12-08
>
> We sincerely thank the reviewer for your prestigious time and valuable feedback! We are encouraged by the positive comments, including “achieves strong performance”, “even with limited data”, and “The experiments are comprehensive and demonstrate the effectiveness”. We will add multiple comments for different aspects to address the reviewer's concerns and questions.

---

> > ### Author Response · Authors · 2025-12-08
> > **Clarification on Our Contribution in Genuine Reasoning**
> >
> > We thank the reviewer for their insightful comments. Our contribution primarily lies in proposing a general approach for metadata-driven structure discovery, applicable to both classic and newly introduced networks.
> >
> > In specific, we highlight results on the bnRep datasets, which are far less likely to appear in any LLM pre-training corpus, as the bnRep package was created in early 2025, while the model (o3) we were using has a training cutoff of Jun 01, 2024. For example, on blockchain, PromptBN (0-sample) achieves an SHD of 15.4, close to 13.2 from the classical HC method using observational samples. Likewise, on consequenceCovid, PromptBN reaches an SHD of 45.0, comparable to 32.6 from HC. PromptBN’s competitive performance suggests that LLMs perform genuine reasoning from metadata rather than relying on prior exposure to any specific graph.
> >
> > To further examine this issue, we conducted targeted contamination tests, as described below.

---

> > ### Author Response · Authors · 2025-12-08
> > **Contamination Tests: Experimental Results and Analysis**
> >
> > We acknowledge that some classic bnlearn benchmarks may appear in LLM pre-training corpora. Since the training data of modern LLMs are undisclosed, contamination cannot be ruled out. **To better understand whether PromptBN’s results may be attributed to memorization rather than reasoning, we conducted three controlled experiments:**
> >
> > * VarName: Variable-name only in the same PromptBN prompt schema
> > * VarDescription: Variable-description only in the same PromptBN prompt schema
> > * Scrambled node index (random permutation) with full metadata in PromptBN
> >
> > Results for all classic networks are provided in Table 3 of the revised manuscript and are summarized below:
> >
> > The results have been added to our Table 3 in the revised paper. The results show that either in VarName or VarDescription setting, the model fails to recover the classical benchmarks well, even with the same variable naming or description. In the scrambled order, however, our PromptBN’s performance is not jeopardized on most of the datasets.
> >
> > | Constraints            | Asia SHD↓ | Asia NHD↓ | Cancer SHD↓ | Cancer NHD↓ | Child SHD↓ | Child NHD↓ | Insurance SHD↓ | Insurance NHD↓ | Alarm SHD↓ | Alarm NHD↓ | Hailfinder SHD↓ | Hailfinder NHD↓ |
> > |----------------------|-----------|-----------|--------------|--------------|------------|-------------|------------------|------------------|-------------|-------------|-------------------|-------------------|
> > | VarName        | 0.0       | 0.000     | 0.67         | 0.030        | 27.3      | 0.050       | 41.3            | 0.050            | 43.8       | 0.030       | 118.67            | 0.040            |
> > | VarDescription | 0.0       | 0.000     | 0.33         | 0.010        | 31.3      | 0.070       | 44.3            | 0.060            | 43.8       | 0.030       | 119.83            | 0.040            |
> > | ScrambleOrder        | 0.0       | 0.000     | 0.6          | 0.024        | 24.8       | 0.051       | 37.4             | 0.044            | 48.4        | 0.964       | 97.8              | 0.032            |
> >
> > Note that we do observe decreased performance on the Hailfinder in the permutation run. We believe it is highly likely due to the common “lost-in-the-middle” phenomenon when the context is quite long. The Hailfinder is the largest dataset we have (56 nodes, 66 edges), and its metadata is lengthy.

---

> > ### Author Response · Authors · 2025-12-08
> > **Complexity: In-depth Analysis and Scalability Discussion**
> >
> > We thank the reviewer for the insightful comments on computational complexity and scalability. **In the revised manuscript, we added a dedicated section that formally analyzes time complexity, LLM query complexity, and scalability for all methods (Section 6).**
> > Updated Table 3 has been added to summarize all time and query complexities, offering a transparent comparison and preventing any overstatement of scalability claims:
> > | **Algorithm**         | **Category** | **Time Complexity**                 | **Query Complexity**              |
> > |-----------------------|--------------|-------------------------------------|-----------------------------------|
> > | Scalability           | LLM          | –                                   | $$\mathcal{O}(1)^* \text{ to } \mathcal{O}(N^2)$$ |
> > | PairwisePrompt        | LLM          | –                                   | $$\mathcal{O}(N^2)$$              |
> > | BFSPrompt             | LLM          | –                                   | $$\mathcal{O}(N)$$                |
> > | ChatPC                | LLM          | $$\mathcal{O}(N^2 \cdot 2^k)$$      | –                                 |
> > | **PromptBN (Ours)**   | LLM          | –                                   | $$\mathcal{O}(1)$$                |
> > | PC-Stable             | Data         | $$\mathcal{O}(N^2 \cdot 2^k)$$      | –                                 |
> > | HC                    | Data         | $$\mathcal{O}(N^2)$$                | –                                 |
> > | GES                   | Data         | $$\mathcal{O}(N^2)^* \text{ to } \mathcal{O}(N^3)$$ | –                   |
> > | LLM-CD(HC)            | Hybrid       | $$\mathcal{O}(N^2)$$                | $$\mathcal{O}(1)^* \text{ to } \mathcal{O}(N^2)$$ |
> > | **ReActBN (Ours)**    | Hybrid       | $$\mathcal{O}(N^2)$$                | $$\mathcal{O}(1)$$                |
> >
> > \* the value represents the best case depending on an internal step’s LLM generation.
> >
> > The key clarifications are as follows:
> >
> > **ReActBN offline runtime.**  We now explicitly distinguish LLM query complexity from CPU-side computation. ReActBN enumerates $\mathcal{O}(N^{2})$ local moves per refinement step. Although the theoretical cost is $\mathcal{O}(T N^{2})$, the number of iterations $T$ is fixed to a small constant in practice, yielding an effective runtime of $\mathcal{O}(N^{2})$. This is comparable to other hybrid methods, such as LLM-CD, which also inherit the complexity of classical search algorithms.
> >
> > **Constant query complexity”** We clarify that this refers strictly to the number of LLM API calls and does not imply constant wall-clock time. ReActBN still performs $\mathcal{O}(T N^{2})$ offline computations, and PromptBN processes metadata whose token length scales with the number of variables. These nuances are now explicitly stated.
> >
> > **Token complexity** We analyzed our token usage using the same tokenizer as the o3 model. The results are shown below (PromptBN):
> >
> > | Dataset    | Variables | Total Tokens | Tokens per Variable | Metadata % |
> > |------------|-----------|--------------|---------------------|------------|
> > | cancer     | 5         | 618          | 32.0                | 25.9%      |
> > | asia       | 8         | 702          | 30.5                | 34.8%      |
> > | child      | 20        | 1,098        | 32.0                | 58.3%      |
> > | insurance  | 27        | 1,322        | 32.0                | 65.4%      |
> > | alarm      | 37        | 1,642        | 32.0                | 72.1%      |
> > | hailfinder | 56        | 2,250        | 32.0                | 79.6%      |
> >
> > For PromptBN, the token count varies only with metadata length, which won’t be unpredictably lengthy. For ReActBN, token usage depends on the (constant) number of iterations and (constant) number of top scores, ensuring scalability across graph sizes.
> >
> > Regarding the monetary cost, since commercial LLM pricing is essentially linear in the number of tokens processed, the monetary cost of PromptBN and ReActBN scales proportionally with the token counts reported above (for PromptBN) and the fixed per-query prompt size times the constant number of iterations (for ReActBN).
> >
> > **Scalability concerns about our methods** In PromptBN, the largest dataset hailfinder only has 2.2k tokens. Most of the modern LLMs have a context window size of 128k or above. Therefore, we believe the scalability of PromptBN is well expected. ReActBN uses a fixed template and injects a constant number of scores into the prompt. Therefore, **the prompt length remains effectively constant**, and scalability w.r.t. LLM query size is not an issue.
> >
> > Potential concerns instead arise on the **CPU side**. Here, ReActBN relies on the same portable scoring function (BIC) widely used in classical methods such as HC. ReActBN introduces no additional computational overhead** beyond the neighborhood enumeration that these methods already perform. Consequently, as the dataset size increases, ReActBN remains feasible whenever Hill-Climbing (HC) itself is feasible**.

---

### Author Response · Authors · 2025-12-08
**Common response to all reviewers**

We thank all reviewers for their time and constructive feedback. In response, we conducted substantial additional work to strengthen the rigor and completeness of the work.

Key revisions include:
* **Comprehensive baseline reproduction and comparative analysis** (Section 5)
*  **Detailed complexity analysis**, covering both offline computation and LLM query complexity, along with an expanded scalability discussion (Section 6)

We have highlighted the revised sections for ease of review and would greatly appreciate your re-evaluation. To ensure full transparency and reproducibility, we have also **open-sourced the complete codebase**, including scripts for all reproduced baselines:
[https://github.com/anonymous-4-review-only/my-anonymous-project-001](https://github.com/anonymous-4-review-only/my-anonymous-project-001)

We hope these extensive updates clearly demonstrate the rigor and authenticity of our work.

---

### Decision · Action_Editor_Z6zF · 2026-01-03

**Recommendation:** Accept with minor revision

**Additional Comments:**

Please fully address the minor issues raised by Reviewer 5trM in the rebuttal.

**Audience:**

Yes

**Audience Explanation:**

The paper explores how LLMs could be leveraged in structure discovery in both data-free and data-aware settings and shows promising results in a wide range of tasks. Since this is an emerging topic lies among sub-communities like causal discovery, Bayesian networks, and LLM reasoning, it will certainly draw attention from TMLR's audiences.

**Claims And Evidence:**

Yes

**Claims Explanation:**

The paper proposes PromptBN and ReActBN to discovery structures of Bayesian networks without and with observational data, respectively. Experiments on a wide range of tasks with different sized graphs show promising results.

Several major concerns raised by reviewers, like data contamination, the lack of detailed complexity analysis, missing baselines, and scalability, have been successfully addressed during the rebuttal.

Overall, the claims are sufficiently supported.